

# Polarization upgrade of specMACS: calibration and characterization of the 2D RGB polarization resolving cameras

Anna Weber[1], Tobias Kölling[1,a], Veronika Pörtge[1], Andreas Baumgartner[2], Clemens Rammeloo[2,b], Tobias Zinner[1], and Bernhard Mayer[1]

[1]Meteorologisches Institut, Ludwig-Maximilians-Universität München, Munich, Germany
[2]Deutsches Zentrum für Luft und Raumfahrt, Institut für Methodik der Fernerkundung, Oberpfaffenhofen, Germany
[a]now at: Max Planck Institute for Meteorology, Hamburg, Germany
[b]now at: National Physical Laboratory, Teddington, United Kingdom

**Correspondence:** Anna Weber (Weber.Ann@physik.uni-muenchen.de)

**Abstract.** The spectrometer of the Munich Aerosol Cloud Scanner (specMACS) is a high spatial resolution hyperspectral and polarized imaging system. It is operated in a nadir-looking perspective aboard the German High Altitude and LOng range research aircraft (HALO) and is mainly used for the remote sensing of clouds. In 2019, its two hyperspectral line-cameras which are sensitive to the wavelength range between 400 and 2500nm were complemented by two 2D RGB polarization resolving cameras. The polarization resolving cameras have a large field of view and allow for multi-angle polarimetric imaging with high angular and spatial resolution. This paper introduces the polarization resolving cameras and provides a full characterization and calibration of them. We performed a geometric calibration and georeferencing of the two cameras. In addition, a radiometric calibration using laboratory calibration measurements was carried out. The radiometric calibration includes the characterization of dark signal, linearity, and noise as well as the measurement of the spectral response functions, a polarization calibration, vignetting correction, and absolute radiometric calibration. With the calibration, georeferenced absolute calibrated Stokes vectors rotated into the scattering plane can be computed from raw data. We validated the calibration results by comparing observations of the sunglint, which is a known target, with radiative transfer simulations of the sunglint.

## 1 Introduction

The remote sensing of clouds and aerosols with polarization measurements has been a very active field of research over the past years. Polarized radiance has the advantage that it is dominated by single scattering (Hansen, 1971) and the contribution from multiple scattering is filtered out. Hence, retrievals based on polarization measurements are less influenced by 3D radiative effects compared to conventional spectral approaches. Multi-angle polarimetric observations allow for improved retrievals and contain additional polarization information which can be used for new retrievals. For example, Goloub et al. (2000) and Riedi et al. (2010) developed retrievals of cloud thermodynamic phase from multi-angle polarimetric observations. Moreover, ice crystal asymmetry was derived from polarization by van Diedenhoven et al. (2013) and retrievals of cloud droplet size distribution from cloudbow observations have been developed by Bréon and Goloub (1998), Alexandrov et al. (2012), McBride



et al. (2020), and Pörtge et al. (2023) to name a few examples for retrievals of cloud properties from polarization measurements. Besides that, polarization measurements have also been used to derive aerosol properties (Dubovik et al., 2019).

There is a number of spaceborne and airborne remote sensing instruments with polarization capabilites such as the POLDER
instrument (Deschamps et al., 1994), RSP (Cairns et al., 1999), AirHARP (Martins et al., 2018), SPEX airborne (Smit et al., 2019), and AirMSPI (Diner et al., 2013) which have successfully applied various polarization based retrievals. The specMACS instrument originally consisted of two hyperspectral cameras in the visible and near-infrared wavelength range (Ewald et al., 2016). Data of the hyperspectral cameras has for example been used to retrieve profiles of cloud droplet effective radius (Ewald et al., 2019) and cloud geometry from oxygen-A-band observations (Zinner et al., 2019). Before the EUREC[4]A field campaign
(Stevens et al., 2021) in 2019 they were complemented by two 2D RGB polarization resolving cameras with a large field of view and high angular and spatial resolution. With that, specMACS became a hyperspectral and polarized imaging system. Hyperspectral and polarization resolving cameras are operated on the same platform allowing for combined and improved retrievals of cloud properties. In this paper, the polarization resolving cameras will be introduced.

In general, any digital imaging sensor has imperfections and non-uniformities due to the manufacturing and sensor elec-
tronics, which have to be assessed and characterized by calibration. In addition, an absolute calibration is necessary for certain retrievals. The polarization resolving cameras of specMACS can be classified as division-of-focal-plane polarimeters. There is a variety of different calibration techniques for division-of-focal-plane polarimeters, which have been reviewed by Giménez et al. (2019) and Giménez et al. (2020). Lane et al. (2022) for example calibrated the monochrome version of the polarization resolving cameras from the same manufacturer as our cameras, however they did not provide an absolute calibration.
Rodriguez et al. (2022) calibrated a camera with the same sensor, but the assumptions they made are not applicable to the specific instrument setup of specMACS, since the specMACS setup includes not only lenses but also a window in front of the cameras. We performed a complete characterization and calibration of the polarization resolving cameras with a geometric calibration as well as a radiometric calibration. The radiometric calibration includes a dark signal and noise characterization, a linearity analysis, vignetting correction, polarization calibration, spectral calibration, and absolute radiometric calibration.
With that, we can compute georeferenced, absolute calibrated Stokes vectors, which are rotated into the scattering plane. Here, the scattering plane is the plane containing the direction of the incoming solar radiation and the viewing direction for every pixel. We completed the calibration measurements at the Calibration Home Base (DLR Remote Sensing Technology Institute, 2016) and present our calibration methods and results in this paper. Finally, we applied the calibration results to measurement data of the sunglint, which is formed by the reflection of sunlight on water surfaces. The sunglint is a known target, so that
we could validate the calibration results by comparing the calibrated measurements with radiative transfer simulations of the sunglint.

The paper is organized as follows. First, an instrument description is given, followed by the geometric calibration methods and results in section 3 and the radiometric calibration of the polarization resolving cameras in section 4. In section 5, the calibration is applied to measurement data and compared with radiative transfer simulations of the sunglint in order to validate
the calibration. Finally, the results are summed up.





## 2 Instrument description

The spectrometer of the Munich Aerosol Cloud Scanner (specMACS) is a hyperspectral and polarized imaging system developed and operated by the Meteorological Insitute of the Ludwig-Maximilians-Universität München (Ewald et al., 2016). It is mainly used for the remote sensing of cloud macro- and microphysical properties.

Originally, specMACS consisted of two hyperspectral cameras, so-called VNIR and SWIR, which are sensitive to the wavelength range between 400nm to 1000nm and 1000nm to 2500nm, respectively. Both cameras were characterized and calibrated by Ewald et al. (2016) for the first time in 2014. Together with the calibration of the polarization resolving cameras in 2021, the calibration of the VNIR camera was repeated and showed no changes beyond the measurement uncertainties. The calibration measurements of the SWIR camera could not be repeated in 2021 since the camera had to be sent to the manufacturer for
repair.

    In the past, specMACS was operated in a ground-based setup, before it was integrated into the German research aircraft HALO (Krautstrunk and Giez, 2012) first looking through the aircraft side window. Since 2016, specMACS is operated in a nadir-looking perspective in the rear of the fuselage of the HALO research aircraft. For that, the cameras were mounted into a pressurized housing with temperature stabilization and humidity control, and a 2cm thick quartz glass window (Heraeus
Herasil 102) in front of the cameras.

    In 2019 for the EUREC$^4$A field campaign (Stevens et al., 2021), the hyperspectral cameras were complemented by two 2D RGB polarization resolving cameras, so-called polLL and polLR (polarization camera looking to the Lower Left and Lower Right relative to flight direction, previously called polA and polB in Pörtge et al. (2023)). Both cameras are LUCID vision Phoenix 5.0 MP Polarization Model cameras (LUCID Vision Labs Inc., 2023) with Sony's IMX250MYR sensor (Sony
Semiconductor Solutions Corporation, 2023). They measure synchronized with an acquisition frequency of 8Hz and have an auto-exposure control system similiar to the one described in Ewald et al. (2016). The sensor of a single camera has 2448 × 2048 pixels. It comprises a combination of a color filter array and a polarizer filter array and can be classified as a division-of-focal-plane polarimeter. The color filter array consists of a Bayer pattern of red, green, and blue color channels and the polarizer filter array consists of four different on-chip directional polarizers with 0, 45, 90 and 135° polarization direction. A
block of 4 × 4 pixels forms a super-pixel, whose pixel layout is visualized in Fig. 1. A super-pixel can be subdivided into blocks of 2 × 2 pixels for each color with the four different polarizers on each pixel. The on-chip directional polarizers are placed below the on-chip micro-lenses to reduce the distance between the polarizers and the photodiodes. With this specific sensor layout, simultaneous measurements of 0, 45, 90, and 135° polarization directions are possible and Stokes vectors can be calculated. The Stokes vector provides a full characterization of electromagnetic radiaton and a quantitative description of
polarization. It is defined as (e.g. Hansen and Travis, 1974)

$$\boldsymbol{S} = \begin{pmatrix} I \\ Q \\ U \\ V \end{pmatrix} = \begin{pmatrix} I_0 + I_{90} \\ I_0 - I_{90} \\ I_{45} - I_{135} \\ I_{\text{left-handed}} - I_{\text{right-handed}} \end{pmatrix}. \tag{1}$$





| r90° | r45° | g90° | g45° |
|------|------|------|------|
| r135° | r0° | g135° | g0° |
| G90° | G45° | b90° | b45° |
| G135° | G0° | b135° | b0° |

**Figure 1.** Pixel layout of a super-pixel with the color and polarization filter array.

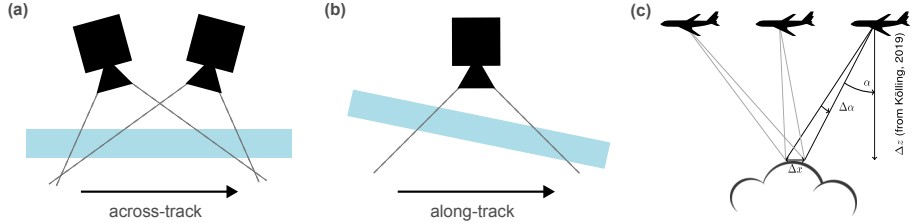

**Figure 2.** (a) and (b) Installation geometry of the two polarizaton resolving cameras. The window in front of the cameras is shown in light blue, the dashed lines indicate the field of view of the cameras. (c) Observation geometry (Figure from Pörtge et al. (2023)).

The polarization resolving cameras can measure the $I$, $Q$, and $U$ component of the Stokes vector. $I$ is the total intensity and $Q$ and $U$ specify linear polarization. $V$ describes circular polarization and is negligible in the atmosphere (e.g. Hansen and Travis, 1974; Emde et al., 2015). The disadvantage of the filter array is that the measurements suffer from sparsity and instantaneous field of view errors. These errors can however be reduced by applying interpolation strategies (Ratliff et al., 2009; Tyo et al., 2009; Gao and Gruev, 2011).

Each of the cameras is combined with a Cinegon 1.8/4.8 lens by Schneider-Kreuznach. The aperture is optimized for the operation on board the HALO research aircraft and set to a fixed value of 5.6. The two polarization resolving cameras are installed with partially overlapping field of views as shown in Fig. 2. This allows for a large overall field of view without the distortions of a fish-eye camera. A single camera has a field of view of $91°$ in along-track and $78°$ in across-track direction and the maximum combined field of view is about $91°$ along-track and $117°$ across-track. The window in front of the cameras is tilted by $12°$ in flight direction relative to the base plate of the instrument (see Fig. 2). A detailed description of the camera geometry is given in the following section. The specifications of the polarization resolving cameras are summarized in Table 1. Example measurements of the polarization resolving cameras are shown in Pörtge et al. (2023).

Applications of the data of the two polarization resolving cameras so far are the stereographic retrieval of 3D cloud geometry (Kölling et al., 2019, does not use polarization information) and the retrieval of cloud droplet size distribution from polarized





**Table 1.** Specifications of the polarization cameras.

| Lenses | Cinegon 1.8/4.8 by Schneider-Kreuznach |
|---|---|
| Focal length | 4.8mm |
| Aperture | 1.8 - 8 (set to 5.6) |
| Along track field of view | 91° |
| Across track field of view | 78° |
| Cameras | Phoenix 5.0 MP Polarization Model |
| Sensor | Sony IMX250MYR CMOS |
| Shutter type | Global |
| Sensor resolution | 5.0 MP |
| Sensor pixels | 2448 × 2048 |
| Pixel size | 3.45µm × 3.45µm |
| Maximum frame rate | 22Hz |
| Bit depth | 12 bit (scaled to 16 bit) |
| Dynamic range maximum | 65535DN |
| Combined field of view | 91° × 117° |

observations of the cloudbow (Pörtge et al., 2023). While flying above a scene, certain (cloud) targets are sampled from different viewing angles (see Fig. 2(c), figure from Pörtge et al. (2023)). This allows for the stereographic reconstruction of the target locations in three dimensional space from which the 3D cloud geometry is derived. But it also provides multi-angle

polarimetric information which can be used for retrievals like the derivation of cloud droplet size distribution by Pörtge et al. (2023).

## 3 Geometric calibration

Both applications of the polarization resolving cameras need an accuracte geometric calibration for the correct localization of the targets. The geometric calibration consists of two steps. First, the camera model has to be defined and camera intrinsics

and distortion coefficients have to be determined. Second, the exact location and orientation of the cameras in a fixed 3D world coordinate system have to be found in order to obtain georeferenced data.

The camera model describes the transformation from world coordinates to pixel coordinates. It relates every pixel to its viewing direction including distortions, e.g. due to lenses along the optical path. The parameters of the camera model include camera intrinsic parameters, distortion coefficients, and extrinsic parameters and can for example be computed with the chess-

board calibration method (Zhang, 2000; Heikkila and Silven, 1997). Multiple views of known targets like the corners of a



chessboard can be used to fit the camera model and solve for the model parameters. Chessboard corners are the intersections of straight lines that are easily detectable and allow the model to be fitted up to subpixel accurary.

We performed the geometric camera calibration using the openCV library (Bradski, 2000) and proceeded similarly to Kölling et al. (2019). But, we applied openCV's rational camera model instead of the thin prism model. In total, we used 249 images of a chessboard with $9 \times 6$ corners with $65\text{mm} \times 65\text{mm}$ square size on an aluminium composite panel for the polLL camera and 212 images of the same chessboard for the polLR camera. The images were taken such that the chessboard corners were distributed over the entire field of view. All measurements were done with the cameras assembled inside the housing with the window in front of them as during aircraft operation. Due to the large field of view of the cameras and the consequential large incident angles on the window and its thickness of 2cm, the window introduces a shift of the viewing directions. This shift results in an additional angle- and distance-dependent distortion of the detected chessboard corners. It cannot easily be included in the camera model since it does not change the direction itself. In order to reduce the impact of the shift, we took the chessboard images with the chessboard in a few meters distance to the instrument - as far away as possible to minimize the impact of the window, but close enough to detect all corners correctly. The mean root mean square reprojection error of the best fit camera model of the polLL and polLR camera amounted to 0.18 und 0.20 pixels, respectively. The field of view of a single camera evaluated to $91° \times 78°$ (along-track $\times$ across-track) and the maximum combined field of view to $91° \times 117°$ corresponding to a maximum combined swath of about 20km$\times$33km at a typical flight altitude of 10km. In addition, the mean angular resolution is $0.04°$ in along-track and across-track direction or about 10m at a typical flight altitude of 10km and a target at ground height. With the acquisition frequency of 8Hz, the polarization resolving cameras provide data with high angular resolution for angular sampling at a typical flight speed of 200m/s which corresponds to an angular resolution of up to $0.14°$ for a target at 10km distance.

In the second step, the camera position and orientation for geoereferencing had to be determined. Precise information about aircraft position and attitude is available from the Basic HALO Measurement and Data System (BAHAMAS). BAHAMAS includes an Inertial Measurement Unit, which is GPS-referenced with data from a Global Navigation Satellite System (Giez et al., 2021). The data is acquired with a rate of 100Hz. After post-processing, the accuracy of the BAHAMAS data is 5m for position data, $0.003°$ for roll and pitch angles and $0.007°$ for true heading (Giez et al., 2021). However, the BAHAMAS sensor is located at the front part of the aircraft, while the specMACS instrument is integrated in the boilerroom in the rear part. Because of that, we expect the accuracy of the BAHAMAS attitude data for the specMACS instrument to be reduced because of bending and stretching of the aircraft fuselage during a flight. Orientation and position of the polarization resolving cameras relative to the aircraft were determined with the method described by Kölling (2020). An initial guess of camera position and orientation is provided from the design documents. We then optimized the rotation angles by projecting specMACS measurements onto satellite images and matching features such as coastlines, lakes, or big roads. This was done once for every measurement campaign since the instrument could be slightly disaligned after each integration into the aircraft. We optimized the orientation angles up to differences of $0.05°$ which corresponds to a shift of 8.7m at the ground for a flight altitude of 10km.





## 4 Radiometric characterization

Besides the geometric calibration, we also characterized the cameras radiometrically. The output of each pixel is given as a digital number (DN). In order to convert this digital number into an absolute radiometric signal, the sensor needs to be calibrated. This calibration includes the investigation of inter-pixel variations due to imperfections of the sensor material as well as other influences from the sensor electronics and optical components. In general, the sensor signal $S$ can be expressed as

$$S = S_0 + S_{\mathrm{d}} + \mathcal{N} \tag{2}$$

with the radiometric signal $S_0$, the dark signal of the sensor $S_{\mathrm{d}}$, and the sensor noise $\mathcal{N}$ (Ewald et al., 2016). The different components of the sensor signal will be characterized in the following sections. The calibration measurements were performed at the Calibration Home Base (CHB; DLR Remote Sensing Technology Institute, 2016; Gege et al., 2009) of the Remote Sensing Technology Insitute of the German Aerospace Center (DLR) in Oberpfaffenhofen in November 2021. All measurements were taken with the cameras mounted inside the housing as during aircraft operation. For radiometric measurements, we used the Large Integrating Sphere (LIS) of the CHB, which is suited for the calibration of instruments with a large field of view. The LIS has a diameter of $1.65\mathrm{m}$, an exit port of up to $55\mathrm{cm}$, and its intensity can be changed by using different combinations of its 18 different lamps. Only pixels iluminated by the lower hemisphere of the LIS were included in the analysis of the radiometric calibration data, because the coating of the lower hemisphere was newer than the coating of the upper hemisphere and due to the large field of view of the cameras, the edge between both hemispheres was visible in the calibration data. If not stated otherwise, all properties are given pixel-wise. Angle brackets denote a temporal average and spatial averages are indicated by an overbar.

### 4.1 Dark signal

The dark signal $S_{\mathrm{d}}$ is a pixel-dependent offset signal the sensor measures when no light penetrates the camera. It can directly be measured from an averaged dark frame $\langle S \rangle = \langle S_0 + S_{\mathrm{d}} + \mathcal{N} \rangle$ since $S_0 = 0$ if no light enters the camera and $\langle \mathcal{N} \rangle \rightarrow 0$. The dark signal can be split into two components and is generally dependent on exposure time $t_{\mathrm{exp}}$ and temperature $T$:

$$S_{\mathrm{d}}(T) = i_{\mathrm{dc}}(T) t_{\mathrm{exp}} + S_{\mathrm{read}}. \tag{3}$$

The dark current $i_{\mathrm{dc}}$ is caused by thermally generated electrons whose generation rate increases with increasing temperature. The read-out offset $S_{\mathrm{read}}$ originates from the A/D converters within the sensor. In contrast to the hyperspectral cameras, the polarization resolving cameras do not have external shutters, which means that no dark signal can be characterized during measurement periods. Because of that, we estimate the dark signal for any measurement during field campaigns from the laboratory characterization.

For the analysis of the spatial structure of the dark signal, we averaged in total 5000 dark frames. The measurements were taken with an exposure time of $5\mathrm{ms}$ and at constant temperature. Results of this analysis are shown in Fig. 3. Mean and





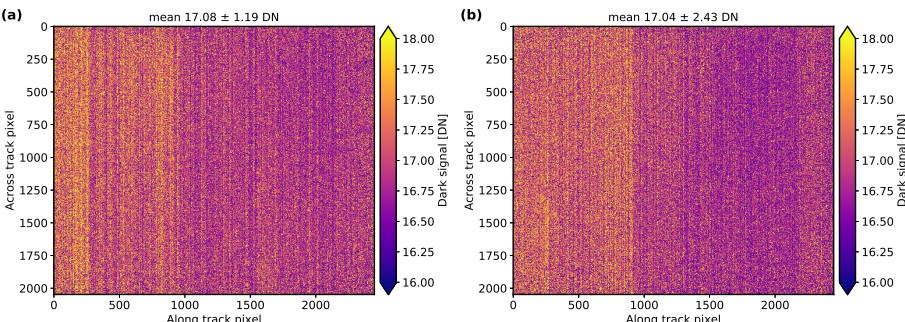

**Figure 3.** Spatial distribution of dark signal averaged over 100 dark measurements with 50 frames each at an exposure time of 5ms and constant temperature for the polLL camera (a) and polLR camera (b).

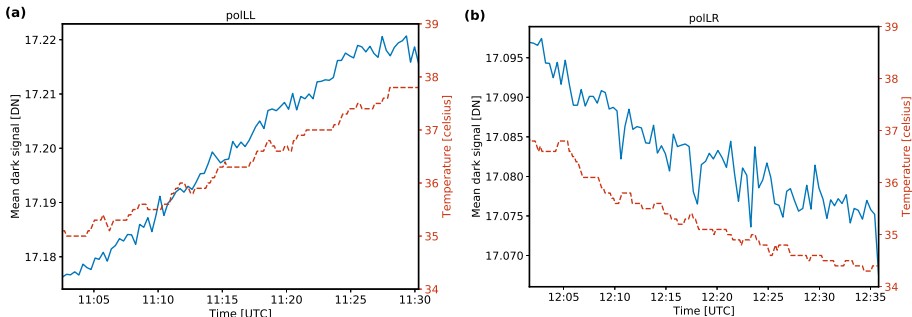

**Figure 4.** Temperature dependence of dark signal averaged over 50 frames for each temperature at an exposure time of 5ms for the polLL camera (a) and polLR camera (b). The red dashed curves indicate the temperature variations, the blue solid curve shows mean dark signal.

standard deviations of the dark signal across the sensor pixels of the polLL and polLR camera amounted to $17.08 \pm 1.19$DN and $17.04 \pm 2.43$DN, respectively. These dark signal levels correspond to 0.026% of the digital number range, which has a dynamic range maximum of 65535DN. For typical signal levels of 30000DN the dark signal acounts for 0.057% of the total signal. In addition, we investigated the temperature dependence of the dark signal (see Fig. 4). During two measurement series the temperature varied between about 35 to 38°C and 34 to 37°C. The temperature was measured by the data logger of the instrument inside the housing and is used as a proxy for the temperature of the sensors. From the analysis, we found dark signal drifts of 0.016 and 0.012 DN/K for the polLL and polLR camera, respectively. Temperature variations during a research flight are usually greatest during take-off and amount to up to 10K until the aircraft reaches its cruising level and the temperature is stabilized at 25°C for the remainder of the flight. Thus, the total dark signal drift due to temperature variations during a research flight is 0.16DN and 0.12DN for polLL and polLR. Lastly, we analyzed the dependence of dark signal on exposure time. We performed dark signal measurements by averaging over 50 frames for exposure times between 0.05 and 100ms at constant temperature. Fig. 5 shows slightly increasing dark signal levels with exposure time. The total dark signal drift due to varying exposure time is 0.23 and 0.22DN for the polLL and polLR camera. Typical exposure times during aircraft



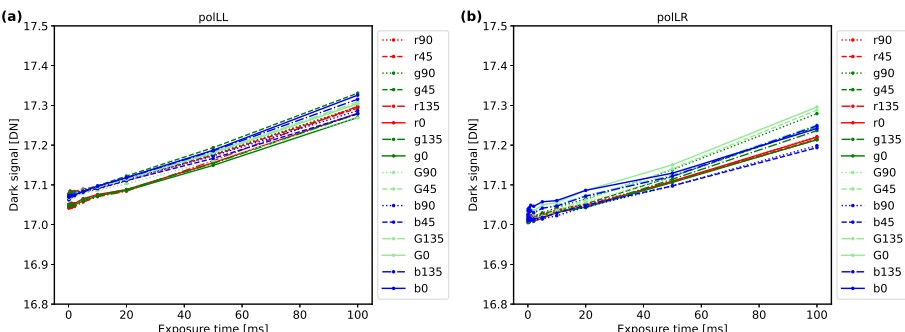

**Figure 5.** Exposure time dependence of dark signal averaged over 50 frames at constant temperature for the polLL camera (a) and polLR camera (b). The different lines show the different color and polarization channels.

operation are below 10ms. In summary, the dark signal level of both cameras is very small with only minor spatial variations and variations with exposure time, and negligible temperature dependence. Thus, we use constant values of 17.08 and 17.04DN

for all channels of the polLL and polLR camera, respectively, for the dark signal correction. The total standard deviation of the dark signal including spatial variations, temperature variations, and variations with exposure time is 1.22 and 2.44DN for polLL and polLR. For typical signal levels of 30000DN, the total dark signal drift corresponds to 0.004% and 0.008% of the signal.

## 4.2 Linearity

Furthermore, we investigated the linearity of the sensors following Forster et al. (2020). According to Ewald et al. (2016), the radiometric signal measured by a perfectly linear sensor with absolute radiometric response $R$ should depend linearly on input radiance $L$ and exposure time $t_{\text{exp}}$:

$$\widetilde{S_0} = RLt_{\text{exp}} = s_{\text{n}}t_{\text{exp}} \tag{4}$$

with the normalized signal or photo current $s_{\text{n}} = RL$. A deviation between the observed signal $S_0$ and the signal expected from

a perfectly linear sensor $\widetilde{S_0}$ is known as photo response non-linearity. To examine the linearity of the polarization resolving cameras, we took measurements with different exposure times above the large integrating sphere assuming that the exposure time is linear. We measured 1000 frames with an acquisition rate of 8Hz for each exposure time and averaged them. The output of the LIS has a standard deviation $\sigma = 0.02\%$ over a time period of 330s and can thus be considered temporally stable for the duration of the measurements (Baumgartner, 2013). We analyzed the linearity of the signal with exposure time for two

different output intensities of the LIS in order to cover a large range of exposure times. The second intensity was about 12% of the first intensity. Fig. 6 shows average radiometric signal $\langle S_0 \rangle$ as a function of exposure time for all different channels of the polLL and polLR camera for both intensities. The mean deviation of the observed signal from the perfectly linear signal of an ideal sensor was determined from a linear fit to the data. Its values are given for every channel in the figure and are generally larger for the measurements at smaller exposure times and higher LIS intensity (panel (a) and (c)). The mean deviations across



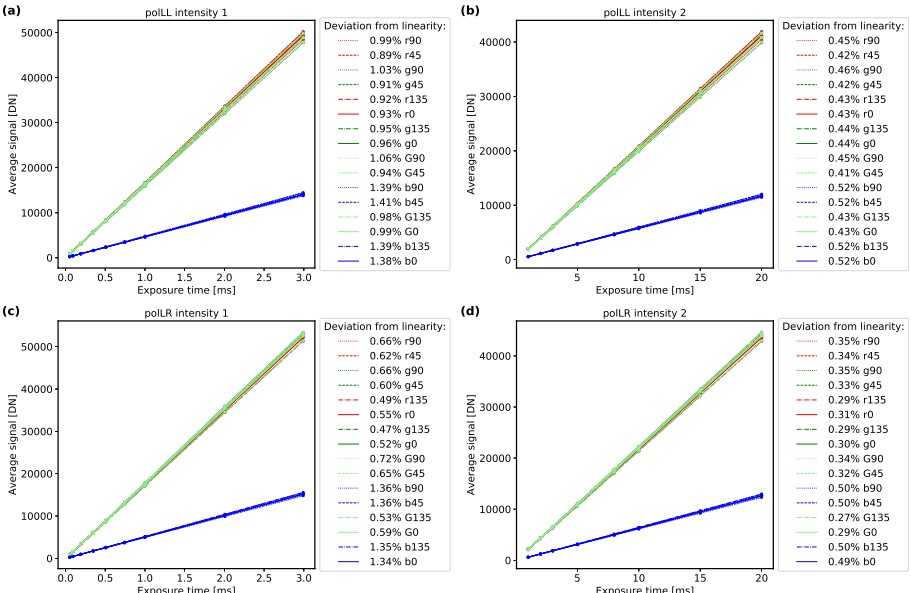

**Figure 6.** Linearity of mean radiometric signal $S_0$ with exposure time for the different channels of the polLL camera ((a) and (b)) and the polLR camera ((c) and (d)) for two different intensities of the large integrating sphere. The dots indicate the observed signal, the lines are a linear fit assuming a perfectly linear sensor. The deviation of the observed signal of each channel from the linear model is given in the legends.

all pixels of a certain color are 0.68%, 0.70%, and 0.96% for the red, green, and blue color channel of the polLL camera and 0.45%, 0.45%, and 0.93% for the polLR camera.

### 4.3 Noise

Noise in general consists of signal noise (photon shot noise) and dark noise. Dark noise is, analog to the dark signal, composed of dark current noise due to statistical fluctuations of thermally generated electrons and read noise from the electronic read-out

process. Photon shot noise originates from the temporally random distribution of photons arriving at the detector. The number of photons measured during a certain time intervall can be described by a Poisson distribution. The standard deviation of a Poisson distribution with expectation value $N$ is proportional to $\sqrt{N}$. Thus, the photon shot noise is directly proportional to the square root of the number $N$ of photoelectrons and the conversion gain $k$, and the total noise $\sigma_{\mathcal{N}}$ can be written as

$$\sigma_{\mathcal{N}} = \sqrt{\sigma_{\text{shot}}^2 + \sigma_{\text{r}}^2} = \sqrt{k^2 N + \sigma_{\text{r}}^2} \tag{5}$$

where $\sigma_{\text{shot}}$ is the photon shot noise and $\sigma_{\text{r}}$ the read noise (Janesick, 2007). For a linear sensor, the measured signal is directly proportional to the number of photons $N$. To analyze the noise characteristics of the polarization resolving cameras, we computed the pixel-wise standard deviation across 1000 frames, which were taken above the LIS for different exposure times and two different output intensities of the LIS in order to cover a large enough signal range. Then, we calculated two-dimensional





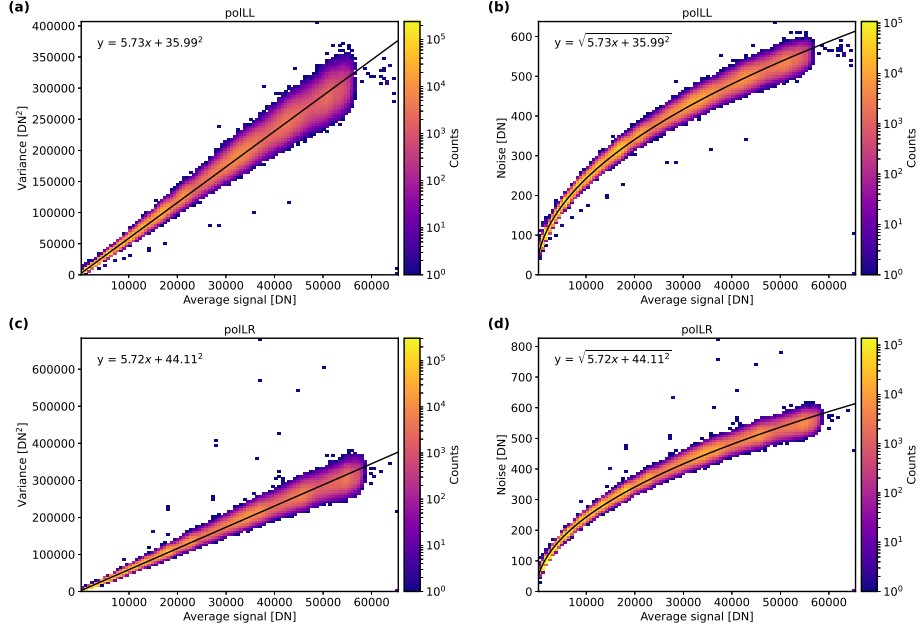

**Figure 7.** Noise characteristics of the r90 channel of the polLL camera ((a) and (b)) and the polLR camera ((c) and (d)). Panels (a) and (c) show two-dimensional histograms of the variance as a function of average dark signal corrected signal and panels (b) and (d) the noise. The solid black lines denote the least-squares fit of the Poisson model given by the equation on the respective panel.

histograms of variance and noise as a function of averaged, dark signal corrected signal and fitted the Poisson model to it. This was done separately for every color and polarization channel. Fig. 7 displays the results for the r90 channel of the polLL and polLR camera. For a typical signal level of 30000DN the noise is around 400DN or 1.3%. All other channels show similar results. The noise characteristics of both cameras are well captured by the Poisson model. Deviations from the Poisson model would be an indication of sensor non-linearities or non-Poisson noise.

## 4.4 Spectral response

The polarization resolving cameras have a color filter array with red, green, and blue color channels. Measurements of the spectral response functions were performed using the Oriel MS257 monochromator of the CHB in the monochromator setup. The wavelength uncertainty of the monochromator is $\pm 0.1$nm in the relevant wavelength range with a spectral bandwidth smaller than $0.54$nm and a relative radiometric uncertainty between 0.6% and 0.9% (Baumgartner, 2019, 2022). We performed measurements for wavelengths between 370nm to 750nm in steps of 5nm at an exposure time of 5ms for both cameras. At every wavelength 50 frames were taken and averaged. Since the monochromator illuminates only a few pixels (about $8 \times 8$), we performed measurements at seven different positions in across track and three different positions in along track direction in order to cover a certain number of pixels distributed across the sensor. We substracted dark measurements from the data, corrected for the monochromator intensity, and normalized the spectral response functions to 1. Fig. 8 shows the resulting mean



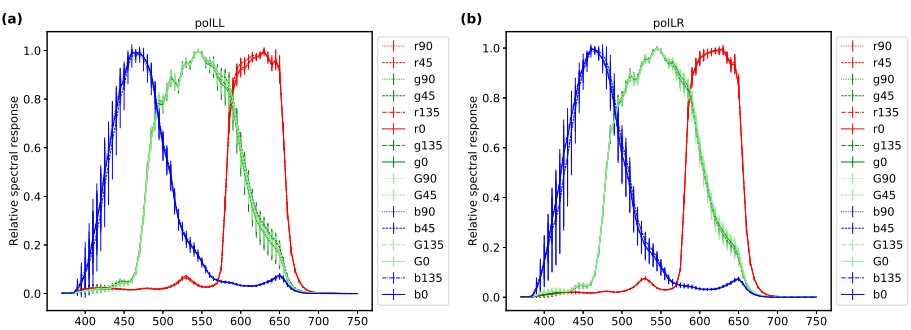

**Figure 8.** Relative spectral response function with uncertainty for the different color and polarization channels of the polLL camera (a) and polLR camera (b).

spectral response functions across all measured pixels for every channel of the polLL and polLR camera. The uncertainties
include the standard deviation across the different pixels and the monochromator uncertainties. In general, there are only
very small differences between the spectral response functions for the different polarization directions of each color. Center
wavelengths and full width at half maximum (FWHM) were determined from a Gaussian fit for the red, green1, green2, and
blue color channels. We obtained values of 621nm (FWHM = 66nm), 548nm (118nm), 545nm (117nm), 468nm (87nm) for
the center wavelengths of the polLL camera and 621nm (67nm), 548nm (119nm), 545nm (117nm), 468nm (87nm) for polLR.

**4.5   Polarization calibration**

In addition to general digital imaging errors, polarimeters have additional inaccuracies due to imperfections of the polarizers.
Transmission, diattenuation, and orientation of the single on-chip directional polarizers can vary across the sensor due to mis-
alignments and non-uniformities from the manufacturing. These variations have to be compensated by polarization calibration
in order to correctly reconstruct the polarization signal from the measurements. There is a variety of different calibration tech-
niques such as single-pixel calibration (Powell and Gruev, 2013), super-pixel calibration (Powell and Gruev, 2013; Lane et al.,
2022) and more complex calibration techniques based on super-pixel calibration (e.g. Chen et al., 2015; Zhang et al., 2016).
Giménez et al. (2019) and Giménez et al. (2020) reviewed different calibration methods for division-of-focal-plane polarime-
ters. They found that the super-pixel calibration method performs well and more complex calibration methods do not improve
the calibration results significantly. Hence, we also applied the super-pixel calibration method. In general, the transformation
of an incident Stokes vector $\boldsymbol{S} = (I, Q, U)^T$ into the measured intensities $\boldsymbol{I} = (I_0, I_{45}, I_{90}, I_{135})^T$ can be described by

$$\boldsymbol{I} = \boldsymbol{AS} + \boldsymbol{d} \tag{6}$$

where $\boldsymbol{d} = (d_0, d_{45}, d_{90}, d_{135})^T$ is the dark signal, which was already characterized in Section 4.1 and $\boldsymbol{A}$ is the so-called transfer
matrix. The aim of the polarization calibration is to determine the transfer matrix $\boldsymbol{A}$. We did this following two approaches.
First, we introduced a theoretical polarization camera model for the transfer matrices for the entire sensor. Second, we used
laboratory calibration measurements to compute transfer matrices and validate the model.



Since the super-pixel method combines pixels with all four different polarizers in a $2 \times 2$ pixels neighborhood, it suffers from instantaneous field of view errors. However, these errors can be reduced by interpolation (Ratliff et al., 2009; Tyo et al., 2009; Gao and Gruev, 2011). Thus, we interpolated the measurements using a bilinear interpolation method in order to obtain measurements of all four polarization directions and all colors at every pixel before we applied the super-pixel calibration to 270 every pixel. To avoid artifacts from extrapolation we excluded the outermost super-pixel. The bilinear interpolation allows for very fast data analysis. It performs well in scenes without sharp edges, which we typically encounter in our application to the remote sensing of clouds. In the future, improved interpolation methods adapted to the combination of color and polarization filter arrays like the methods by Mihoubi et al. (2018) or Morimatsu et al. (2020) could be applied.

### 4.5.1 Theoretical polarization camera model

First, we present the theoretical polarization camera model. The transfer matrix of an ideal sensor follows from equation 1:

$$
\boldsymbol{A}_{\text{ideal}} = \frac{1}{2} \begin{pmatrix} 1 & 1 & 0 \\ 1 & 0 & 1 \\ 1 & -1 & 0 \\ 1 & 0 & -1 \end{pmatrix}.
\tag{7}
$$

Manufacturing imperfections lead to a deviation of the pixel transfer matrices from the ideal matrix and to variations between the pixels. Lane et al. (2022) calibrated the monochromatic version of the polarization resolving cameras from the same manufacturer and found that the transfer matrices are consistent across the sensor and a single matrix can be applied to all 280 pixels. In addition, the deviation between the measured matrices and the ideal matrix was small.

The polarization resolving cameras of specMACS are integrated into a housing with a window. This window can change the polarization state of the light passing through it and therefore must be considered in the polarization calibration. Its impact on the polarization can be described by the Mueller matrix of a linear diattenuator $M_{\text{window}}$, which is given in Bass et al. (2010). The matrix depends on the incident angles on the window as well as the refractive index of the window and is computed for 285 every pixel and every color. The refractive index for every color is obtained by integrating the wavelength dependent refractive index of the window with the spectral response functions of the cameras. Moreover, the incident angles can be calculated using the geometric calibration. Fig. 9 shows the 1,1- and 1,2-components of $M_{\text{window}}$ for the red channel of both cameras. The 1,1-component describes the total transmission through the window, which is slightly reduced towards the corners with increasing incident angles on the window. $M_{\text{window, 1,2}}$ specifies the impact of the window on the polarization.

In total, the transfer matrix is calculated with

$$
\boldsymbol{A} = \boldsymbol{A}_{\text{ideal}} \boldsymbol{M}_{\text{window}}.
\tag{8}
$$

This transfer matrix does not include the lenses in front of the cameras. Since the lenses are in reality a combination of lenses whose exact design is not known to us, it was not possible to include a theoretical Mueller matrix for the lenses as well. However, according to Lane et al. (2022), the choice of the camera lens has only little influence on the transfer matrices.




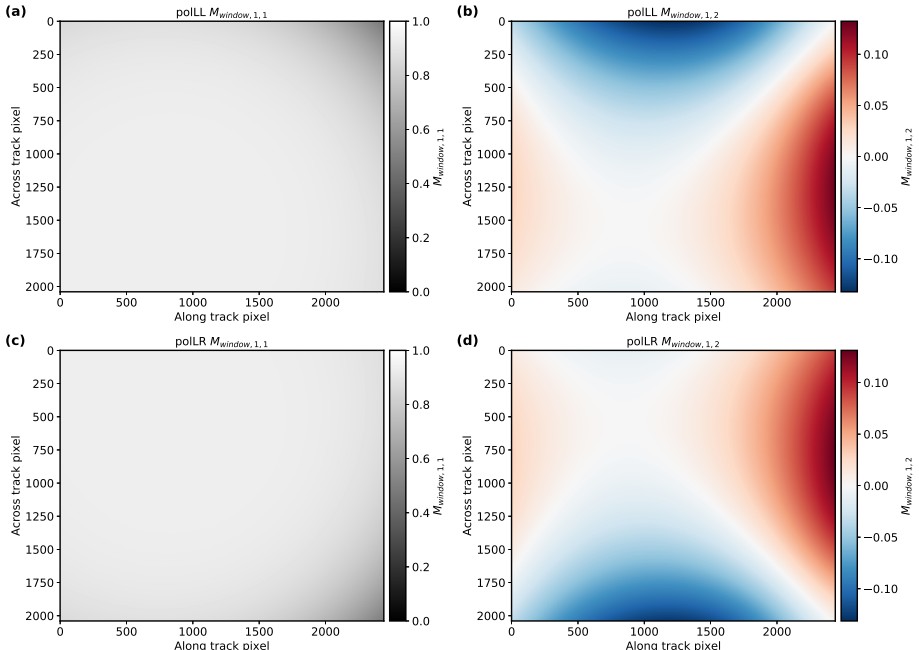

**Figure 9.** 1,1- and 1,2-components of the Mueller matrix describing the window for the red channel of the polLL and polLR camera.

Because of that, we assume that our theoretical model of the transfer matrices is a good approximation. Nevertheless, we validated the theoretical model with a laboratory polarization calibration.

### 4.5.2  Laboratory polarization calibration

For the laboratory polarization calibration, we performed calibration measurements with the polarizer setup of the CHB. A linear polarizer (Moxtek UBB01A) was mounted at a motorized rotation stage between specMACS and the large integrating

sphere. This rotatable polarizer has a contrast ratio larger than 1000 for incident angles up to $\pm 20°$. With respect to the measurement precision of the polarization cameras, we thus considered the calibration light source as perfectly linearly polarized. We rotated the polarizer from -180° to 180° in steps of 15° and took 50 frames per angle which we averaged. Due to the large field of view of the polarization resolving cameras and the limited size of the polarizer only a small part of the sensor was iluminated by the LIS with the polarizer. In order to cover at least large parts of the field of view, we tilted the entire instrument

and took measurements for 32 different tilt angles in along-track and across-track track direction and in nadir direction. Consequently, we achieved polarization measurements for 28.8% of the pixels of the polLL camera and 29.6% of the pixels of the polLR camera. Because of the size and the large weight of the specMACS instrument including the complete aircraft housing and environment control, tilting the instrument is very difficult, and it was not possible to obtain measurements for the entire field of view. The intensity of the sphere was monitored and stayed constant during the measurements.





We evaluated the polarization calibration measurements using the super-pixel calibration method. Since the output of the LIS after passing the polarizer was not known, we used normalized quantities similiar to Lane et al. (2022):

$$\boldsymbol{I}_n - \boldsymbol{d}_n = \boldsymbol{A}\boldsymbol{S}_n \tag{9}$$

with the normalized intensities $\boldsymbol{I}_n = \frac{2}{I_0 + I_{45} + I_{90} + I_{135}}\boldsymbol{I}$, the normalized dark signal $\boldsymbol{d}_n = \frac{2}{I_0 + I_{45} + I_{90} + I_{135}}\boldsymbol{d}$, and the normalized incoming Stokes vector

$$\boldsymbol{S}_n = \begin{pmatrix} 1 \\ \cos 2\phi \\ \sin 2\phi \end{pmatrix}. \tag{10}$$

Here, $\phi$ is the polarization angle of the rotatable polarizer. With that, $\boldsymbol{A}$ could be fitted from the measured dark signal $\boldsymbol{d}_n$ and intensities $\boldsymbol{I}_n$ for every pixel using equation 9.

The Stokes vector as well as the transfer matrix are always defined relative to a reference plane. In connection with the polarization calibration, we distinguish three different reference systems. The laboratory reference system is defined by the plane containing the $0°$-axis of the motorized rotation stage and the normal of the polarizer. Moreover, the reference plane for the camera reference system for each camera is given by the $x$-$z$-plane of the camera coordinate system with the $x$-axis parallel to the direction of the $0°$ polarizer and the $z$-axis normal to the focal plane array. Finally, the Stokes vectors can be rotated from the camera reference system into the scattering plane. The transformation from the camera coordinate system to the scattering plane is known from the geometric calibration and varies between different observation geometries. Thus, with the laboratory polarization calibration, we aim for computing the transfer matrices in the camera reference system.

For that, we defined the polarizer angle $\phi$ for the incoming Stokes vector $\boldsymbol{S}_n$ relative to the $0°$-axis of the motorized rotation stage and computed the transfer matrices first in the laboratory reference frame with the normalized super-pixel method described above. Therefore, we combined the laboratory measurements for different tilt angles into one laboratory reference system and solved for the transfer matrices. We only included illuminated pixels with viewing directions within $\pm 20°$ perpendicular to the polarizer where the polarizer can be considered perfect. In addition, we excluded pixels with dirt or reflections on the window.

In a second step, we transformed the obtained transfer matrices from the laboratory reference system into the camera reference system. The direct determination of the rotation from the laboratory to the camera reference frame through the identification of the polarizer orientation visible in the measurements was not possible due to the angle dependent shift introduced by the window, which is relevant at small distances. However, for single scattering, the $U$ component of the Stokes vector is zero in the scattering plane due to symmetries. We used this fact to find the rotation from the laboratory to the camera reference frame using measurements taken during the EUREC[4]A campaign (Stevens et al., 2021) by minimizing $U$ along the scattering plane. Contributions from multiple scattering can in principle cause deviations of $U$ from zero. To minimize the influence of multiple scattering, we chose measurements from EUREC[4]A without clouds and minimum amount of aerosol. We applied the computed transfer matrices in the laboratory reference frame to measurement data from the EUREC[4]A campaign and rotated the obtained Stokes vectors with a single rotation matrix first into the camera coordinate system and next for every pixel from



the camera coordinate system into the scattering plane. Since the transformation from the camera coordinate system to the scattering plane is known we could optimize for the rotation from the laboratory to the camera coordinate system by minimizing the absolute value of $U$ along the scattering plane. With that, we obtained transfer matrices in the camera reference system for every measured pixel.

Mean and standard deviation of $A$ across all measured sensor pixels for the red channel of the polLL and polLR camera are

$$A_{\text{polLL,red}} = \frac{1}{2}\begin{pmatrix} 0.988 & 0.972 & 0.012 \\ 1.010 & -0.021 & 0.986 \\ 0.991 & -0.976 & -0.014 \\ 1.006 & 0.025 & -0.984 \end{pmatrix} \pm \begin{pmatrix} 0.012 & 0.008 & 0.058 \\ 0.005 & 0.059 & 0.007 \\ 0.012 & 0.005 & 0.057 \\ 0.006 & 0.060 & 0.007 \end{pmatrix} \tag{11}$$

and

$$A_{\text{polLR,red}} = \frac{1}{2}\begin{pmatrix} 0.989 & 0.972 & -0.019 \\ 1.007 & 0.019 & 0.993 \\ 0.991 & -0.978 & 0.016 \\ 1.010 & -0.012 & -0.991 \end{pmatrix} \pm \begin{pmatrix} 0.010 & 0.008 & 0.057 \\ 0.005 & 0.060 & 0.007 \\ 0.010 & 0.005 & 0.059 \\ 0.005 & 0.059 & 0.007 \end{pmatrix}. \tag{12}$$

Other channels gave similar results. These transfer matrices include the impact of the entire optical system on the polarization including the window and the lenses. Since we used normalized intensities, they do not contain variations of absolute transmission across the sensor and absolute radiometric pixel response non-uniformity which can be calibrated separately with a flat-field calibration. The deviation of the computed transfer matrices from the ideal matrix is small, as is the standard deviation of the transfer matrices across the sensor pixels. This indicates that the assumption of the ideal transfer matrix in the theoretical polarization model does not introduce large calibration errors.

A first quality check of the super-pixel polarization calibration is the reconstruction error which is the relative deviation of the reconstructed Stokes vectors using the measured intensities and the computed inverse transfer matrix from the incoming Stokes vectors. The mean reconstruction error across all pixels was smaller than $10^{-13}\%$ for the $I$ component of the Stokes vector for both cameras. For the $Q$ component it amounted to $(-0.37 \pm 0.60)\%$ for the red channel of the polLL camera, $(-0.25 \pm 0.42)\%$ for the red channel of the polLR camera, and values in the same order of magnitude for the other color channels of both cameras. In addition, we computed the relative calibration error introduced by Lane et al. (2022) for the red, green, and blue color channels of the polLL and polLR cameras. The error is defined as

$$Err = \frac{2}{\sqrt{3}}\|A - A_{\text{ideal}}\|_F \tag{13}$$

where $\|\|_F$ is the Frobenious norm. It gives the upper limit of the error which is made when a polarization resolving camera is used uncalibrated by applying the ideal transfer matrix and if the light is totally linearly polarized. For partially polarized light the error is smaller (Lane et al., 2022). The relative calibration error of the mean transfer matrix for the red, green, and blue color channel amounts to 3.5%, 3.5%, and 4.6% for the polLL camera and 3.1%, 3.1%, and 4.0% for the polLR camera.

The polarization calibration error when using the theoretical polarization camera model introduced in the previous section is expected to be even smaller since the model additionally includes the window. Moreover, polarization measurements of





clouds are usually only partially polarized leading to a reduced relative polarization calibration error. Hence, the polarization calibration results from the theoretical model covering the entire field of view will be used in the following.

## 4.6 Vignetting correction

Vignetting describes the intensity fall-off from pixels in the center towards pixels in the edges of the sensor. On the one hand, the brightness of off-axis image points is naturally reduced due to the geometry of the optical system. On the other hand, 375 optical vignetting is caused by optical components like lenses. Off-axis incident light is blocked by physical components like the aperture and the edge of a lens leading to an intensity decrease for rays with larger angles towards the sensor edges (Gross, 2011; Bass et al., 2010). Vignetting can be corrected for by applying a flat-field model $F$, which approximates the vignetting functions. The flat-field corrected signal can be computed from the radiometric signal with

$$S_{\text{F}} = S_0/F. \tag{14}$$

380 We used the parabolic vignetting model by Kordecki et al. (2016), defined as

$$F = a_{\text{x}}x^2 + b_{\text{x}}x + a_{\text{y}}y^2 + b_{\text{y}}y + c \tag{15}$$

since it showed a better agreement with the observed vignetting compared to typical radial models for $F$. Here, $x$ and $y$ are the pixel coordinates. All other parameters have to be determined from measurements of a uniformly illuminated scene. For that, we performed flat-field measurements using the LIS. We computed Stokes vectors from the dark signal corrected intensities 385 with the transfer matrices from the polarization calibration and used the normalized $I$-component of the Stokes vector to fit the flat-field model for every color channel separately. Again, only the lower hemisphere of the LIS was included in the analysis and pixels with reflections and dirt on the window were excluded. Fig. 10 and 11 display the results of the red channel for the polLL and polLR camera, respectively. All other channels showed a similar behaviour. The mean deviation between the model and the measurements for the red, green, and blue channel is $(0.0 \pm 1.2)\%$, $(0.0 \pm 1.3)\%$, and $(0.0 \pm 1.4)\%$ for polLL 390 and $(0.0 \pm 1.3)\%$, $(-0.1 \pm 1.3)\%$, and $(0.0 \pm 1.3)\%$ for polLR.

## 4.7 Absolute radiometric response

Finally, the dark signal corrected, exposure time normalized and flat-field corrected Stokes vectors have to be converted into absolute radiances. In general, the absolute radiance $L$ in $\text{mW m}^{-2}\text{nm}^{-1}\text{sr}^{-1}$ is computed from the normalized signal $s_{\text{n}}$ in DN $\text{s}^{-1}$ with the absolute radiometric response $R$ (Ewald et al., 2016):

395 $$L = R^{-1} \cdot s_{\text{n}}. \tag{16}$$

Here, the normalized signal is given by the exposure time normalized and vignetting corrected Stokes vector

$$s_{\text{n}} = S_0/(Ft_{\text{exp}}) \tag{17}$$

with $S_0 = A^{-1}(I - d)$. In order to determine the absolute radiometric response, we again used measurements of the LIS. We averaged 1000 frames and computed the exposure time normalized, vignetting corrected Stokes vectors. The measured output



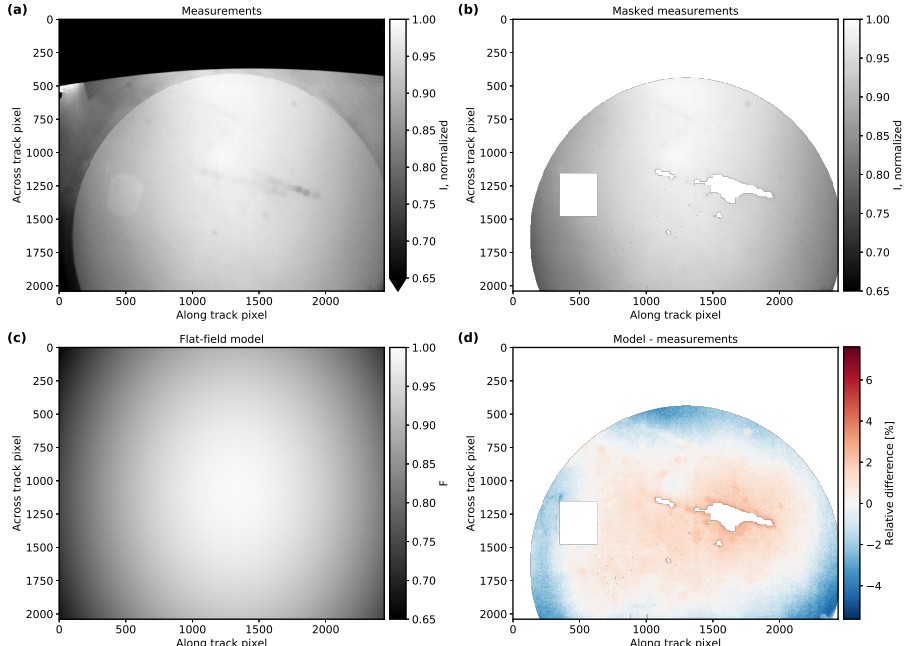

**Figure 10.** Flat-field model results for the red channel of the polLL camera. (a) Mean normalized, dark signal corrected total intensity. (b) Part of measurement in (a) used in the vignetting correction (removed LIS upper hemisphere, reflections, and dirt affected areas). (c) Vignetting model fitted to the measurements. (d) Relative difference between vignetting model and measurements.

400 spectrum of the LIS was integrated with the spectral response functions to obtain radiance values $L$ for every color channel. Then, we computed the absolute radiometric response $R$ using only the $I$ component of the normalized Stokes vectors $s_{n,0}$

$$R = \frac{s_{n,0}}{L}. \tag{18}$$

Assuming that the photo response non-uniformity had already been accounted for by applying the vignetting correction, we computed a single absolute radiometric response for all pixels of a color channel by taking the mean across all pixels of

405 the channel. Table 2 summarizes the resulting values of the absolute radiometric response $R$ for the red, green, and blue color channels of the polLL and polLR camera. The uncertainties include the standard deviation of $R$ across all pixels, the uncertainty of the output of the LIS (Rammeloo and Baumgartner, 2023) as well as the spatial nonuniformity of the LIS, and the uncertainty of the spectral response functions. The photo response non-uniformity remaining after the vignetting correction is contained in the standard deviation of $R$ across all pixels. The systematic difference between the absolute radiometric response of the two

410 cameras could come from the manual aperture setting.

In summary, absolute calibrated Stokes vectors in the camera reference system can be calculated from the interpolated measured intensities via

$$\boldsymbol{S} = \mathbf{A}^{-1}(\boldsymbol{I} - \boldsymbol{d})/(RFt_{\exp}). \tag{19}$$



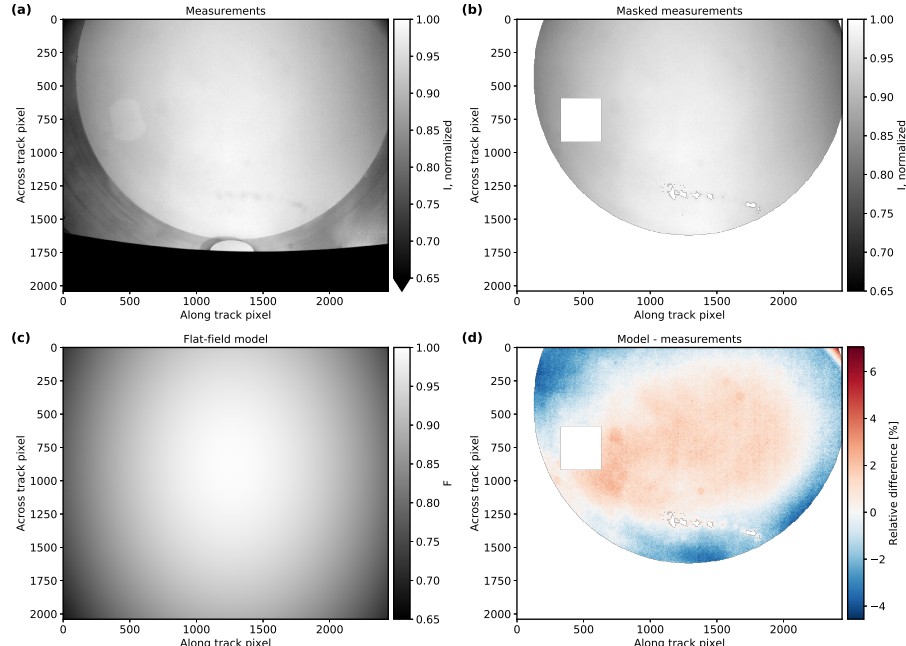

**Figure 11.** Flat-field model results for the red channel of the polLR camera. (a) Mean normalized, dark signal corrected total intensity. (b) Part of measurement in (a) used in the vignetting correction (removed LIS upper hemisphere, reflections, and dirt affected areas). (c) Vignetting model fitted to the measurements. (d) Relative difference between vignetting model and measurements.

**Table 2.** Absolute radiometric response $R$ for the different color channels of the polLL and polLR camera in DN s$^{-1}$/(mW m$^{-2}$nm$^{-1}$sr$^{-1}$) with uncertainties.

|        | red             | green            | blue            |
|--------|-----------------|------------------|-----------------|
| polLL  | $44120 \pm 706$ | $60823 \pm 1367$ | $31607 \pm 950$ |
| polLR  | $46549 \pm 766$ | $65471 \pm 1247$ | $34559 \pm 1137$ |

For further application of the measured polarization data to retrievals like the retrieval by Pörtge et al. (2023), the Stokes vector of each pixel is rotated into its scattering plane with the Mueller rotation matrix $\boldsymbol{M}_{\mathrm{rot}}$ (e.g., Mischenko et al., 2002):

$$\boldsymbol{S} = \boldsymbol{M}_{\mathrm{rot}}\boldsymbol{A}^{-1}(\boldsymbol{I} - \boldsymbol{d})/(RFt_{\mathrm{exp}}). \tag{20}$$

## 4.8 Total radiometric uncertainty

The estimation of the total radiometric uncertainty was achieved similarly to Ewald et al. (2016) by applying Gaussian error propagation. The uncertainty of the radiometric signal $S_0$ (here $\boldsymbol{I}_0 = \boldsymbol{I} - \boldsymbol{d}$) is given by the uncertainties of the dark signal $\sigma_{\mathrm{d}}$





and the instantaneous noise of the signal $\sigma_{\mathcal{N}}$:

$$\sigma_{\boldsymbol{I}_0} = \sqrt{\sigma_{\mathrm{d}}(t_{\mathrm{exp}}, T)^2 + \sigma_{\mathcal{N}}^2}. \tag{21}$$

The uncertainty of the dark signal consists of the dark signal drift due to the temperature and exposure time dependence and the standard deviation of the dark signal across all sensor pixels as discussed in Section 4.1. Next, the uncertainty of the normalized and vignetting corrected Stokes vectors can be computed. It consists of the uncertainty of the radiometric signal $\sigma_{\boldsymbol{I}_0}$, the uncertainty due to the sensor nonlinearity, the uncertainty of the polarization calibration and of the vignetting correction. The uncertainty of the polarization calibration is composed of the uncertainty of the transfer matrices $\sigma_{\mathrm{A}}$ and a rotation uncertainty of the Stokes vectors $\sigma_{\mathrm{rot}}$ due to the uncertainty of the geometric calibration when rotating the Stokes vectors into the scattering plane.

$$\frac{\sigma_{\boldsymbol{s}_{\mathrm{n}}}}{\boldsymbol{s}_{\mathrm{n}}} = \sqrt{\left(\frac{\sigma_{\boldsymbol{I}_0}}{\boldsymbol{I}_0}\right)^2 + \left(\frac{\sigma_{\mathrm{nonlin}}}{\boldsymbol{s}_{\mathrm{n}}}\right)^2 + \left(\frac{\sigma_{\mathrm{A}}}{\boldsymbol{s}_{\mathrm{n}}}\right)^2 + \left(\frac{\sigma_{\mathrm{rot}}}{\boldsymbol{s}_{\mathrm{n}}}\right)^2 + \left(\frac{\sigma_F}{F}\right)^2} \tag{22}$$

We estimated the upper limit of the uncertainty of the transfer matrices with the deviation of the laboratory transfer matrices from the ideal transfer matrix using the error defined by Lane et al. (2022). This is a very conservative estimate of the upper limit since we included the impact of the window in the transfer matrices and our transfer matrices are thus more accurate than the ideal transfer matrix alone. Additionally, typical observations of clouds are only partially polarized leading to a smaller relative polarization calibration error. The rotation error is zero for the $I$ component of the Stokes vector, since the total intensity is invariant under rotations, but it is non-zero for $Q$. Finally, the absolute radiometric uncertainty is given by the combination of the uncertainty of the normalized Stokes vectors and the absolute radiometric response:

$$\frac{\sigma_{\boldsymbol{S}}}{\boldsymbol{S}} = \sqrt{\left(\frac{\sigma_{\boldsymbol{s}_{\mathrm{n}}}}{\boldsymbol{s}_{\mathrm{n}}}\right)^2 + \left(\frac{\sigma_{\mathrm{R}}}{R}\right)^2}. \tag{23}$$

The uncertainty estimation was done for every color channel. Typical values of the absolute radiometric uncertainty are given in Table 3. The largest contribution to the total radiometric uncertainty is due to the polarization calibration. In general, total radiometric uncertainty is larger towards the corner regions and smaller in the center of the image. Due to the larger incident angles, the impact of the lenses as well as the window on polarization increases towards the corners leading to a increased uncertainty of the polarization calibration compared to the center region.

## 5 Validation

Finally, we applied the calibration to measurement data to compute georeferenced, absolute calibrated Stokes vectors rotated into the scattering plane. Moreover, the results are compared to simulations in order to validate the calibration. The sunglint originates from specular reflection of sunlight on the rough ocean surface. Observations of the sunglint are very well suited for a validation of the calibration with simulations, since it is a known target. Sunglint observations have for example been used for the in-flight calibration of POLDER (Toubbe et al., 1999). Fig. 12 (a) - (d) shows an example sunglint observation of the



**Table 3.** Absolute radiometric uncertainty $\frac{\sigma_S}{S}$ of the red, green, and blue color channels of the polLL and polLR cameras for a typical signal level of 30000DN and a typical value of the degree of linear polarization $DOLP = 0.5$ as in the cloudbow region.

|       |     | red  | green | blue |
|-------|-----|------|-------|------|
| polLL | $I$ | 4.1% | 4.4%  | 5.7% |
|       | $Q$ | 4.1% | 4.5%  | 5.8% |
| polLR | $I$ | 3.8% | 3.9%  | 5.5% |
|       | $Q$ | 3.8% | 4.0%  | 5.5% |

polLR camera measured on 2020-01-22 at 16:20 UTC west of Barbados above the tropical atlantic ocean during the EUREC[4]A
field campaign. The sunglint is visible as a maximum in the total intensity in panel (a) and minimum in $Q$ in panel (c) around
the specular direction. The $U$ component in panel (d) is much smaller than $Q$ as it is expected for Stokes vectors rotated into
the scattering plane.

We performed polarized simulations of this specific observation with libRadtran (Mayer and Kylling, 2005; Emde et al.,
2016) and the Monte Carlo solver MYSTIC (Mayer, 2009; Emde et al., 2010). For the ocean surface, we used the bidirectional
reflectance distribution function (BRDF) by Cox and Munk (1954a, b), which we extended to polarization by considering
the polarization dependent Fresnel reflectivities. The sunglint shape and maximum intensity depend on wind speed and wind
direction. The wind speed affects the width of the sunglint and can be fitted to measurements. For that, we performed several
simulations for different wind speeds and determined the best fit wind speed by a least-squares fit to the observation data along
the scattering plane. Wind direction was taken from data of the METEOR ship, which was measuring close to the location of the
HALO aircraft at the time of the observation. Since the measurement was taken above the tropical atlantic ocean, we assumed
the tropical maritime aerosol mixture from the OPAC library (Hess et al., 1998) and derived aerosol mass concentrations for
the mixture from data of the WALES lidar (Wirth et al., 2009), which was also measuring on board HALO using the method
by Gutleben (2020). We chose an example observation with only small amounts of aerosol to reduce the uncertainty due to
uncertainties of the retrieval and measurements of aerosol mass concentrations. In order to obtain simulations for the different
color channels, we simulated a spectrum and integrated it with the spectral response functions derived during the laboratory
calibration. To exclude situations with cirrus clouds above the HALO aircraft during the observation, the BACARDI cloud flag
(Ehrlich et al., 2021) was used. An undected cirrus cloud between the sun and the aircraft would lead to a reduced sunglint
intensity and discrepancies between observations and simulations.

Simulation results for the observation in Fig. 12 (a) - (d) are shown in Fig. 12 (e) - (h). In general, simulation and observation
agree well. Both, simulated and observed Stokes vectors are rotated into the scattering plane for comparison. The $Q$ component
is much larger than the $U$ component as it is expected from symmetries. Differences between the observation and the simulation
and their mean and standard deviation across all pixels can be seen in Fig. 12 (i) - (l). The mean absolute difference between the
observation and simulation is smaller than $0.7\mathrm{mW\,m^{-2}nm^{-1}sr^{-1}}$ for all Stokes vector components. Mean relative differences
of the $I$ and $Q$ components are -1.1% and 3.0%, respectively. Thus, the observation and simulation agree within the expected





**Figure 12.** (a) - (d) Example sunglint observation of the green channel of the polLR camera on 2020-01-22 at 16:20 UTC. (e) - (h) Sunglint simulation corresponding to the observation in (a) - (d). (i) - (l) Absolute difference of the sunglint observation and simulation in (a) - (d) and (e) - (h). Mean and standard deviation of the absolute differences are given in the respective titles. (a), (e), (i) Total intensity. (b), (f), (j) Degree of linear polarization, (c), (g), (k) $Q$ component of the Stokes vector. (d), (h), (l) $U$ component of the Stokes vector. The dashed lines indicate scattering angles.

uncertainties. The small scales structures which are visible in the sunglint observations are due to the orientation of single waves on the ocean surface, which, of course, is not represented in the simulation. The $I$ component of the measured Stokes vectors is smaller than the simulations around the sunglint maximum and the $Q$ component in the same region is larger in the



measurements compared to the simulations while the differences between simulated and observed $I$ and $Q$ components are small outside the sunglint, see Fig. 12 (i), (k). This can be explained by the uncertainty of the wind speed and wind direction
for the ocean BRDF in the simulations, since they affect the sunglint maximum intensity. An error in the absolute radiometric calibration would lead to differences across the entire field of view also outside the sunglint. The degree of linear polarizations shows in general only small differences of $0.02 \pm 0.03$ with the largest values towards the corners. Since the degree of linear polarization is independent of the absolute calibration, the deviations between measurements and simulations are due to the uncertainty of the polarization calibration or a deviation of the assumed atmospheric constituents in the simulation to the
observed ones which affect the polarization in the simulations. The differences could for example be explained by a small polarization impact of the lens in front of the camera or of the on-chip microlenses, which are not included in the theoretical polarization calibration. A more accurate laboratory calibration of the entire field of view including the corners would be necessary to quantifiy their impact. Due to the size and weight of the instrument, it was not possible to take calibration measurements for the corner regions with the setup at the CHB. In addition, deviations of the assumed aerosol properties from
the measured ones could have an impact. The larger deviations of $U$ from zero in the observations compared to the simulations can as well be explained by the uncertainty of the polarization calibration and the aerosol properties with the polarization calibration being the dominant factor. In addition, uncertainties of the geometric calibration lead to uncertainties of the rotation into the scattering plane which could cause deviations from zero. However in summary, the observation and simulation agree within the expected uncertainties. For more accurate results, a laboratory polarization calibration for the entire field of view is
necessary to include the polarization impact of all optical components. In addition to the validation of the laboratory calibration, the sunglint observations and simulations could also be used for an in-flight calibration, which is very useful to continuously monitor the stability of the cameras between laboratory calibrations.

## 6 Summary

In this paper, we introduced the polarization upgrade of specMACS. In 2019, before the EUREC[4]A field campaign, the hy-
perspectral cameras of specMACS were complemented by two 2D RGB polarization resolving cameras. The two polarization resolving cameras have a large combined field of view of about $91° \times 117°$ (along-track $\times$ across-track) and high angular and spatial resolution. We performed a complete calibration and characterization of the polarization resolving cameras and repeated the calibration of the VNIR spectrometer from Ewald et al. (2016). To this end, we conducted calibration measurements at the Calibration Home Base. Concerning the VNIR camera, we did not find significant differences between the calibration from
2016 and the new calibration.

With the calibration of the polarization resolving cameras, we obtain georeferenced, absolute calibrated Stokes vectors rotated into the scattering plane from the raw data. The geometric calibration of the polarization resolving cameras included a chessboard calibration for the determination of the camera model as well as a method for georeferencing. In addition, we completed a radiometric calibration of the two cameras. The dark signal was characterized to account for 0.057% of the signal
for typical signal levels with a small spatial variabiliy across the sensor pixels, exposure time dependency, and temperature

dependency of in total 0.004% and 0.008%. Moreover, the noise characteristics were captured well by the Poisson model and the non-linearity of the sensor was found to be below 1%. Furthermore, we computed the spectral response for every channel from calibration measurements and did a polarization calibration. For the polarization calibration, we used a theoretical camera model which we validated with a laboratory calibration. Flat-field measurements were done and evaluated to obtain a vignetting

correction. Finally, we carried out an absolute radiometric calibration of the cameras and calculated the total radiometric uncertainty, which ranges between 3.8% and 5.8% for the different channels of the two cameras for typical signal levels. The uncertainty is dominated by the uncertainty of the polarization calibration and increases from the center of the sensor towards the corners.

Afterwards, we applied the calibration to measurement data from the EUREC$^4$A campaign and validated it with simulations.

For that, we used observations of the sunglint, which is a well characterized target and compared the observations to polarized radiative transfer simulations of the same measurement scene. This method could also be used for an in-flight calibration of the polarization resolving cameras to continuously monitor the stability of the sensor in between laboratory calibrations. We found agreement between observations and simulations within the characterized accuracy and thus validated our calibration.

*Author contributions.* TK, TZ, and BM realized the polarization upgrade of specMACS. TK developed the data acquisition software for the

polarization resolving cameras, the data file format, and the methods for geometric camera calibration. VP, AB, CR, and AW performed the calibration measurements. AB and CR processed the data of the utilities of the CHB. AW evaluated the calibration data and wrote the manuscript with input from all co-authors.

*Competing interests.* The authors declare that they have no conflict of interest.

*Acknowledgements.* We would like to thank Markus Rapp (Institute of Atmospheric Physics, DLR Oberpfaffenhofen) for financing the

calibration measurements at the Calibration Home Base. In addition, we thank Martin Wirth for providing the WALES lidar data for the sunglint simulation, Stefan Koppenhofer for his support during the laboratory calibration, and Claudia Emde for valuable explanations concerning polarization.



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
