# Peer review of "Polarization upgrade of specMACS: calibration and characterization of the 2D RGB polarization resolving cameras"

_EGUsphere, 2023_

## Referee Comment (RC1)

General comments:

The paper is well written, clear, and of interest. I recommend publication with a few comments.

Specific comments:

- Eq. (1): When measuring the total intensity $I$ with a polarization camera, I think it is preferable to use $I = (I_0 + I_{45} + I_{90} + I_{135})/2$ rather than $I = I_0 + I_{90}$. Ideally of course, it should be that $I_0 + I_{90} = I_{45} + I_{135}$. Nevertheless, I think it is preferable to take the information of all four pixels into account.

- Section 4.5.2 Laboratory polarization calibration:
  I think this section could be improved. I am not sure I fully understand the calibration procedure. Some additional step-by-step explanations with equations and/or a figure explaining the three reference systems and how they relate would help me. To be a bit more specific:
    o In Step 1 (Line 326-331): Did you compute Transfer matrix Eq. (8) by means of Eq. (9) with the camera being in the camera reference frame? Is the result a transfer matrix from laboratory frame (linear polarizer) to camera frame that contains a rotation matrix that still needs to be determined? Or are camera and linear polarizer in the same laboratory reference system?
    o Line 332-345: I assume the problem that is being solved here is finding the rotation induced by the window. So if the Stokes vector is rotated beforehand, does Eq. (9) become something like this,
$$I_n - d_n = A \cdot R \cdot S_n,$$
    with R being the rotation matrix we are looking for?
    If so, why could you not simply fit a misalignment factor dphi similar to Eq. (13) in Lane et al. (2022) (This misalignment factor is also merely a rotation of angle dphi). I understand the sentence spanned from line 333-335, but couldn't you still optimize for the rotation by rotating the linear polarizer? How are the EURECA measurements polarized with respect to the camera reference system (or the scattering plane)? I think it is worth giving more details.

- The statements in Line 279 "a single matrix …. To all pixels" and Line 294 "the camera lens has only little influence" citing Lane et al. are slight oversimplifications. Lane et al. used a 105m lens set to f/22 (fairly straight rays) to show that the super-pixels on the sensor are generally consistent. When they compare the lenses, they merely focus on the central pixels. However, and presumably particularly important for wide-angle lenses, lenses can show an effect called polarization aberration of lenses. This is nicely explained in the reference [1], section 1.7.2, page 22 ff. (also note the effect of high numerical aperture wavefronts described in section 1.7.3). The effect is particularly high at the edges (see Fig. 1.38 in [1]), which might explain your larger differences in the corners (mentioned in line 482). My suggestion would be: it is fair to assume one transfer matrix for all pixels, as the superpixels should generally be consistent across the entire sensor. However, this will probably not fully correct the entire lens (as you already concluded yourselves in line 486). I do not see a need to change any data / results. But it is worth to correct the statements and to mention the potential effect of polarization aberration of lenses.

Technical corrections:

- Eq. (1): It should be $I_{right} - I_{left}$ and not $I_{left} - I_{right}$, see [1], page 64, Eq. (3.1)
- Line 137, 142: altitude instead of attitude

[1]: Chipman, Russell, Wai Sze Tiffany Lam, and Garam Young. *Polarized light and optical systems*. CRC press, 2018.

---

## Referee Comment (RC2)

This paper discusses the calibration pipeline for the polarization resolving cameras of the spectrometer of the Munich Aerosol Scanner (specMACS). The authors discuss the instrument itself, then delve into geometric calibration of the image frame, dark characterization, non-linearity, spectral response, polarization calibration of an enclosing window and optical assembly, flatfielding, and absolute radiometric response. The authors close with a discussion of overall uncertainty and a measurement-model intercomparison over sunglint measured during a recent field campaign.

This paper comes after Portge et al. (2023), which demonstrated the polarimetric cloud retrieval capabilities of the same specMACS cameras over marine and popcorn cumulus cloud fields. Polarimetric remote sensing is a hot topic in the climate community right now. Papers that demonstrate polarimeter instrument calibration (as well as their science) will continue to be relevant, interesting, and useful to AMT readers. I recommend publication with minor/optional revisions.

In-line comments:

Line 85: I understand that the Equation 1 is the form given in Hansen and Travis (1974), though the Sony sensor allows for a more comprehensive calculation using all four angles (shown in Lane et al. 2022):

$$I = \frac{1}{2}(I_0 + I_{90} + I_{45} + I_{135})$$

This form is used later in the paper (as a normalization during polarization calibration, line 313), but it isn't clear to me if the actual intensity measurement (Stokes I) is calculated this way throughout the entire paper. Either way, please harmonize the definition of I across the paper. Also, the typical convention of V in Eq. (1) is flipped from what is shown (right – left).

Section 3. I recommend to add a figure to this section. A visual of the chessboard calibration from the perspective of the specMACS sensor would help a lot with the interpretation here.

Line 193: Figure 3 shows that polLL and polLR have systematic differences in the forward and aft sides of the dark frame. Even at a ~2.5 counts spread, this structure is important to capture, and could be relatively easy to apply in post-processing on image data. At 30000DN, I agree it will not make much of a difference to use a single value or adopt a spatial dark map for correction. However, I imagine part of the interest in specMACS data goes beyond clouds – possibly to science retrievals of aerosol, land, ocean, and free atmosphere properties. Many of these targets will not have 30000DN signals. For example, open ocean is extremely dark in RGB and can go to <5% in DOLP off-glint, like in Figure 12b. At these low light levels, a few counts could be important. Also, the later sections discuss the many ways that the calibration could be improved – using the spatial field of the dark (and scaling the dark counts relative to any measurement temperature) could be a step in this direction. I would consider including this in the calibration pipeline instead of a single value for the dark.

Figure 6. It is challenging to differentiate the curves in each figure due to the overlap and large scale. This could be stronger as a residual plot (i.e. $\widetilde{S_o}$ - $S_o$), as a function exposure time for all pixels shown. Also, please make the points larger.

Line 215: Relative to the detector spec, are these non-linearities reasonable?

Figure 7. Is there is any new information in (a) and (c) that isn't already in (b) and (d)? If not, I recommend to only show (b) and (d).

Figure 8. Though the smaller peaks in blue @ 650nm and in red @ 550nm are typical of some Bayer filter designs, how is this addressed in the radiometric calibration? This could be important for cross-talk considerations, and may have some influence on the error analysis in Table 3.

Line 278 + Line 294: This may not hold for the entire specMACS FOV, though. The Lane et al. (2022) study prioritized pixels near the image center and predicted higher errors in focus and polarization measurement at large AOI. Since a single-camera specMACS FOV is decently large (~45 deg nadir-to-aft), and the Cinegon lens does not seem to be telecentric (from the spec), there will likely be AOI-related differences in the transfer matrix at wider angles. I am glad this is recognized by the authors in the discussion towards the end of the paper, but I would reword these statements to differentiate specMACS from the Lane et al. (2022) study a bit more here.

Line 320: I strongly recommend to add a figure that visually explains these three reference frames. It is difficult to reconcile them from the text alone and the following paragraphs require the reader to fully understand each one.

Line 359: The reconstruction error on I of $10^{-13}$% is incredibly small versus the error on Q. Even with normalized intensities, I would still expect to see a reconstruction error in the ballpark of what is reported for Q. This suggests to me that the derivation of the transfer matrix is weighting the I inputs more strongly than Q or U. Can you give more details on how that value was derived?

Line 365: Why is it useful to know that the specMACS pol cameras would be between 3 and 5% biased, if they were used uncalibrated while imaging a target with DOLP = 1? Most scientists will never use uncalibrated specMACS data – maybe this is a marker of how close the instrument is to an ideal calibration already? Either way, I suggest changing this to how much error we could expect to see in a calibrated specMACS DOLP measurement (or defer this to Table 3 – see comment below).

Line 390: Systematic and spatial differences between model and measurement in Figures 10 and 11 on the order of 2-6% are quite large for a flatfield residual. This may impact science retrievals done in specific pixel regions – was there any reason not to trust the spatial distribution of the LIS field outright? Integrating spheres should be excellent spatial sources for flatfield.

The other way to approach this could be to step the specMACS field of view across the LIS aperture while taking images. This would place the LIS aperture in different locations of the FOV and fully cover the FPA in a "composite" flatfield over all images taken. Was a test like this considered? I am not requesting extra work, but for this section, I would add more details about why a model was preferred despite significant spatial residuals in Figures 10 and 11.

Line 435: I recommend including a table that lists the sigma errors for each of the terms in Eq. (22), for each wavelength – or if some are functions, give the functional form. Much of this data is already given throughout the paper, but a summary table is preferable.

Table 3. Can you also provide the uncertainty for DOLP? This is a benchmark used to gauge the overall polarization accuracy of a multi-angle polarimeter. This may take further propagation of Eq. (22), but it is also important to show (especially relative to typical atmospheric signals).

---

## Author Comment (AC1)

**Reply to referee comment #1**

We thank Referee #1 for reviewing the manuscript and the valuable comments and suggestions which we address below. The responses to the referee comments are given in blue italic letters.

General comments:

The paper is well written, clear, and of interest. I recommend publication with a few comments.

Specific comments:

− Eq. (1): When measuring the total intensity $I$ with a polarization camera, I think it is preferable to use $I = (I_0 + I_{45} + I_{90} + I_{135})/2$ rather than $I = I_0 + I_{90}$. Ideally of course, it should be that $I_0 + I_{90} = I_{45} + I_{135}$. Nevertheless, I think it is preferable to take the information of all four pixels into account.

*Thank you for your comment. We do indeed compute the I-component of the Stokes vector from our measurements with $I = (I_0 + I_{45} + I_{90} + I_{135})/2$ using all four measured intensities. So, it was misleading to give the general definition of the Stokes vector in Eq. (1). We changed the equation to $I = (I_0 + I_{45} + I_{90} + I_{135})/2$ to be consistent.*

− Section 4.5.2 Laboratory polarization calibration: I think this section could be improved. I am not sure I fully understand the calibration procedure. Some additional step-by-step explanations with equations and/or a figure explaining the three reference systems and how they relate would help me. To be a bit more specific:
  o In Step 1 (Line 326-331): Did you compute Transfer matrix Eq. (8) by means of Eq. (9) with the camera being in the camera reference frame? Is the result a transfer matrix from laboratory frame (linear polarizer) to camera frame that contains a rotation matrix that still needs to be determined? Or are camera and linear polarizer in the same laboratory reference system?

    *We added more details about the different reference systems and also tried to describe the first step in more detail to make that clearer. In addition, we added a reference with sketches which visualize the different reference systems. We computed the transfer matrices in this first step in the laboratory reference frame by solving equation 9 in a least-squares sense similarly to Rodriguez et al. (2022). With the resulting transfer matrix Stokes vectors in the laboratory reference frame can be computed from measured intensities. The procedure of the laboratory polarization calibration in section 4.5.2 is independent of the theoretical polarization calibration model (equation 8) in section 4.5.1. Equation 8 gives Stokes vectors in the camera reference system. The transfer matrices obtained in the first step of the laboratory polarization calibration give results in the laboratory reference system and are rotated to the camera reference system in the second step of the laboratory polarization calibration.*

  o Line 332-345: I assume the problem that is being solved here is finding the rotation induced by the window. So if the Stokes vector is rotated beforehand, does Eq. (9) become something like this,

$$I_n - d_n = A \cdot R \cdot S_n,$$

with R being the rotation matrix we are looking for? If so, why could you not simply fit a misalignment factor dphi similar to Eq. (13) in Lane et al. (2022) (This misalignment factor is also merely a rotation of angle dphi). I understand the sentence spanned from line 333-335, but couldn't you still optimize for the rotation by rotating the linear polarizer? How are the EURECA measurements polarized with respect to the camera reference system (or the scattering plane)? I think it is worth giving more details.

*In contrast to Lane et al. (2022) the polarizer in our setup was not mounted on a manual but on a motorized rotation stage. This means that the relative orientation of the polarizer for the different measurements with different rotation angles are very accurate and we did not have to account for misalignments due to manual arrangement. But, what we needed to determine was the absolute orientation of the 0 degree direction of the linear polarizer in camera coordinates. In principle, it would also have been possible to rotate the incoming Stokes vector first into the camera reference system and then determine the transfer matrices as you propose with your equation above. However, we did not have a "ground truth" for the 0 degree direction which we could have used to directly optimize for such a misalignment factor due to window in front of the cameras. Because of that, we used the known property, that U=0 in the scattering plane for single scattering with our measurements from the EUREC4A campaign as described in the paper. The measurements are at first raw data, from which we computed Stokes vectors in the laboratory reference frame with the transfer matrices of step 1. Then we applied two rotation matrices to the Stokes vectors, one for the transformation from laboratory to camera reference system, which had to be optimized, and the second one for transforming from the camera reference system to the scattering plane, which was known from the geometrical calibration. By minimizing U in the scattering plane we could then find the rotation from laboratory to camera reference frame.*
*We added also more details to this section to make our methods more comprehensible.*

*The entire part about the laboratory polarization calibration reads now:*

[revised manuscript text omitted]

− The statements in Line 279 "a single matrix …. To all pixels" and Line 294 "the camera lens has only little influence" citing Lane et al. are slight oversimplifications. Lane et al. used a 105m lens set to f/22 (fairly straight rays) to show that the super-pixels on the sensor are generally consistent. When they compare the lenses, they merely focus on the central pixels. However, and presumably particularly important for wide-angle lenses, lenses can show an effect called polarization aberration of lenses. This is nicely explained in the reference [1], section 1.7.2, page 22 ff. (also note the effect of high numerical aperture wavefronts described in section 1.7.3). The effect is particularly high at the edges (see Fig. 1.38 in [1]), which might explain your larger differences in the corners (mentioned in line 482). My suggestion would be: it is fair to assume one transfer matrix for all pixels, as the superpixels should generally be consistent across the entire sensor. However, this will probably not fully correct the entire lens (as you already concluded yourselves in line 486). I do not see a need to change any data / results. But it is worth to correct the statements and to mention the potential effect of polarization aberration of lenses.

*Thank you very much for pointing that out. We reworded both lines and added more details to make the differences between the setup of specMACS and Lane et al. (2022) clearer and avoid oversimplifications. Lines 279 and 294 read now:*

*"Lane et al. (2022) calibrated the monochromatic version of the polarization resolving cameras from the same manufacturer. They focused on the central pixels of the sensor and found that the transfer matrices are consistent across this sensor region and a single matrix can be applied to all pixels. In addition, the deviation between the measured matrices and the ideal matrix was small for the central pixel region with small incident angles which they considered."*

*And*

*"According to Lane et al. (2022), the choice of the camera lens has only little influence on the transfer matrices for the central pixel region of the camera where the incident angles of the rays are small. Thus, we assume that our theoretical model of the transfer matrices is a good approximation. However, lenses can introduce polarization aberrations especially for larger incident angles towards the corner regions (Chipman et al., 2018). This effect is not included in the theoretical polarization calibration model. Because of that, we validated the theoretical model with a laboratory polarization calibration."*

Technical corrections:

− Eq. (1): It should be $I_{right} - I_{left}$ and not $I_{left} - I_{right}$, see [1], page 64, Eq. (3.1)

 *Thank you very much for noting that. We corrected the equation.*

− Line 137, 142: altitude instead of attitude

 *We do indeed mean the attitude of the aircraft here. The BAHAMAS data provides aircraft position (latitude, longitude, height) and attitude (roll, pitch, and yaw angles). We added more details to clarify that:*
 *"Precise information about aircraft position (latitude, longitude, and altitude of the aircraft) and attitude (roll, pitch, and yaw angles) is available from the Basic HALO Measurement and Data System (BAHAMAS)."*

[1]: Chipman, Russell, Wai Sze Tiffany Lam, and Garam Young. Polarized light and optical systems. CRC press, 2018

---

## Author Comment (AC2)

**Reply to referee comment #2**

We thank Brent McBride for reviewing the manuscript and the valuable comments and suggestions which we address below. The responses to the referee comments are given in blue italic letters.

This paper discusses the calibration pipeline for the polarization resolving cameras of the spectrometer of the Munich Aerosol Scanner (specMACS). The authors discuss the instrument itself, then delve into geometric calibration of the image frame, dark characterization, non-linearity, spectral response, polarization calibration of an enclosing window and optical assembly, flatfielding, and absolute radiometric response. The authors close with a discussion of overall uncertainty and a measurement-model intercomparison over sunglint measured during a recent field campaign. This paper comes after Pörtge et al. (2023), which demonstrated the polarimetric cloud retrieval capabilities of the same specMACS cameras over marine and popcorn cumulus cloud fields. Polarimetric remote sensing is a hot topic in the climate community right now. Papers that demonstrate polarimeter instrument calibration (as well as their science) will continue to be relevant, interesting, and useful to AMT readers. I recommend publication with minor/optional revisions.

In-line comments:

Line 85: I understand that the Equation 1 is the form given in Hansen and Travis (1974), though the Sony sensor allows for a more comprehensive calculation using all four angles (shown in Lane et al. 2022):

$$I = 1/2(I_0 + I_{90} + I_{45} + I_{135})$$

This form is used later in the paper (as a normalization during polarization calibration, line 313), but it isn't clear to me if the actual intensity measurement (Stokes I) is calculated this way throughout the entire paper. Either way, please harmonize the definition of I across the paper. Also, the typical convention of V in Eq. (1) is flipped from what is shown (right – left).

*Thank you for your comment. We do indeed compute the I-component of the Stokes vector from our measurements with $I = (I_0 + I_{45} + I_{90} + I_{135})/2$ using all four measured intensities. So, it was misleading to give the general definition of the Stokes vector in Eq. (1). We changed the equation to $I = (I_0 + I_{45} + I_{90} + I_{135})/2$ to be consistent throughout the paper. In addition, we corrected the equation for the V component. Thank you very much for noting that.*

Section 3. I recommend to add a figure to this section. A visual of the chessboard calibration from the perspective of the specMACS sensor would help a lot with the interpretation here.

*We added a figure showing an example image of a chessboard as recommended.*

Line 193: Figure 3 shows that polLL and polLR have systematic differences in the forward and aft sides of the dark frame. Even at a ~2.5 counts spread, this structure is important to capture, and could be relatively easy to apply in post-processing on image data. At 30000DN, I agree it will not make much of a difference to use a single value or adopt a spatial dark map for correction. However, I imagine part of the interest in specMACS data goes beyond clouds – possibly to science retrievals

of aerosol, land, ocean, and free atmosphere properties. Many of these targets will not have 30000DN signals. For example, open ocean is extremely dark in RGB and can go to <5% in DOLP off-glint, like in Figure 12b. At these low light levels, a few counts could be important. Also, the later sections discuss the many ways that the calibration could be improved – using the spatial field of the dark (and scaling the dark counts relative to any measurement temperature) could be a step in this direction. I would consider including this in the calibration pipeline instead of a single value for the dark.

*Thank you very much for this comment. For our applications of the data to the remote sensing of cloud macro- and microphysical properties, using a single value for the dark signal is accurate enough, which is why we did not use the spatial field of the dark signal. The spread of about 2-3 counts is very small and the temperature dependency of the dark signal of about 0.16 during a typical flight negligible for our applications. But we agree that for other applications this might become relevant. We added a sentence noting that the calibration of the instrument could be improved by using the spatial field instead of single value and will keep that in mind for potential future applications to darker scenes:*
*"Moreover, the spatial field of the dark signal could be used instead of a single value for the dark signal correction to further reduce the calibration uncertainties for retrievals of e.g. aerosol or land properties with very small signal levels."*

Figure 6. It is challenging to differentiate the curves in each figure due to the overlap and large scale. This could be stronger as a residual plot (i.e. $\tilde{S_o}$ - $S_o$), as a function exposure time for all pixels shown. Also, please make the points larger.

*Thank you very much for noting that. We changed the marker and increased the size of the points as suggested to make it easier to differentiate. We are aware that the overlap of the curves in the plot is not ideal. The reason for choosing this visualization is that we wanted to show the linear scaling of the measured signal with exposure time which is not visible in a residual plot. To further quantify the deviation of the signal from the linear relationship we added the percentages in the legend of the plot.*

Line 215: Relative to the detector spec, are these non-linearities reasonable?

*The camera specifications do not include information about non-linearities. However, the non-linearities we found are reasonable compared to other cameras (e.g. Forster et al. 2020).*

Figure 7. Is there is any new information in (a) and (c) that isn't already in (b) and (d)? If not, I recommend to only show (b) and (d).

*According to Poisson statistics the variance scales linearly with the signal. This linear behavior can nicely be seen in panels (a) and (c) which is why we added those two panels. We added this to the text to make it clearer:*
*"The noise characteristics of both cameras are well captured by the Poisson model. Panels (a) and (c) show the expected linear relationship between the variance and the signal while the noise scales with the square root of the signal in panels (b) and (d)."*

Figure 8. Though the smaller peaks in blue @ 650nm and in red @ 550nm are typical of some Bayer filter designs, how is this addressed in the radiometric calibration? This could be important for cross-talk considerations, and may have some influence on the error analysis in Table 3.

*The radiometric calibration was performed at the large integrating sphere whose output spectrum was measured by the CHB. We integrated this spectrum with the previously determined spectral response functions of the three color channels to obtain the output radiance of the LIS for the respective color channels for the absolute radiometric calibration. In this sense the smaller peaks were accounted for.*

Line 278 + Line 294: This may not hold for the entire specMACS FOV, though. The Lane et al. (2022) study prioritized pixels near the image center and predicted higher errors in focus and polarization measurement at large AOI. Since a single-camera specMACS FOV is decently large (~45 deg nadir-to-aft), and the Cinegon lens does not seem to be telecentric (from the spec), there will likely be AOI-related differences in the transfer matrix at wider angles. I am glad this is recognized by the authors in the discussion towards the end of the paper, but I would reword these statements to differentiate specMACS from the Lane et al. (2022) study a bit more here.

*Thank you for this comment. We changed the lines you mentioned and added more details to make the differences between the specMACS setup and Lane et al. (2022) clearer and avoid oversimplifications. Lines 279 and 294 read now:*
*"Lane et al. (2022) calibrated the monochromatic version of the polarization resolving cameras from the same manufacturer. They focused on the central pixels of the sensor and found that the transfer matrices are consistent across this sensor region and a single matrix can be applied to all pixels. In addition, the deviation between the measured matrices and the ideal matrix was small for the central pixel region with small incident angles which they considered."*
*and*
*"According to Lane et al. (2022), the choice of the camera lens has only little influence on the transfer matrices for the central pixel region of the camera where the incident angles of the rays are small. Thus, we assume that our theoretical model of the transfer matrices is a good approximation. However, lenses can introduce polarization aberrations especially for larger incident angles towards the corner regions (Chipman et al., 2018). This effect is not included in the theoretical polarization calibration model. Because of that, we validated the theoretical model with a laboratory polarization calibration."*

Line 320: I strongly recommend to add a figure that visually explains these three reference frames. It is difficult to reconcile them from the text alone and the following paragraphs require the reader to fully understand each one.

*We added more details about the different reference systems in the text and also tried to give more detailed descriptions of the different steps of the laboratory polarization calibration in order to make our methods more comprehensible. In addition, we added a reference which includes sketches visualizing the different reference systems. The section explaining the reference systems reads now:*
*"The Stokes vector as well as the transfer matrix are always defined relative to a reference plane. In connection with the polarization calibration, we distinguish three different reference systems. The laboratory reference system is defined by the plane containing the 0◦-axis of the linear polarizer*

*between the large integrating sphere and the instrument and the normal of this polarizer. Moreover, the reference plane for the camera reference system for each camera is given by the x-z-plane of the camera coordinate system with the x-axis parallel to the 0∘-direction of the polarizers on the sensor and the z-axis normal to the focal plane array of the camera. Finally, the Stokes vectors can be rotated from the camera reference system into the scattering plane. The scattering plane is the plane containing the vector of the incoming solar radiation and the viewing direction of ach pixel. Sketches visualizing the different reference systems can for example be found in Eshelman et al. (2019). The transformation from the camera coordinate system to the scattering plane is known from the geometric calibration and varies between different observation geometries with different vectors of the incoming solar radiation. Thus, with the laboratory polarization calibration, we aim for computing the transfer matrices in the camera reference system."*

Line 359: The reconstruction error on I of 10-13% is incredibly small versus the error on Q. Even with normalized intensities, I would still expect to see a reconstruction error in the ballpark of what is reported for Q. This suggests to me that the derivation of the transfer matrix is weighting the I inputs more strongly than Q or U. Can you give more details on how that value was derived?

*The transfer matrices were derived by solving equation 9 in a least-squares sense with $A = (I_n - d_n)S_n^{-1}$ where $S_n^{-1}$ is the pseudo-inverse of the incoming Stokes vectors $S_n$. The measured Stokes vectors were then reconstructed from the measured intensities via $S_{n,r} = A^{-1} (I_n - d_n)$. Finally, we computed the reconstruction error which we define as the relative difference between the reconstructed Stokes vectors $S_{n,r}$ and the incoming Stokes vectors $S_n$. The given values of the reconstruction error are mean values across all measured pixels. We added an additional reference and more detailed description of the method throughout the entire section and included the equations above to the paragraph.*

Line 365: Why is it useful to know that the specMACS pol cameras would be between 3 and 5% biased, if they were used uncalibrated while imaging a target with DOLP = 1? Most scientists will never use uncalibrated specMACS data – maybe this is a marker of how close the instrument is to an ideal calibration already? Either way, I suggest changing this to how much error we could expect to see in a calibrated specMACS DOLP measurement (or defer this to Table 3 – see comment below).

*With the laboratory polarization calibration, we analyzed the polarization properties and determined transfer matrices for significant parts of the field of view but we could not cover the entire field of view of the cameras. Because of that, we developed the theoretical polarization model and validated it with the laboratory polarization calibration. In this context, the polarization calibration error introduced by Lane et al. (2022) is useful, since it indicates that the instrument is in fact close to an ideal calibration concerning polarization. Thus, the use of the ideal transfer matrix in the theoretical model covering the entire field of view can be justified and introduces only small errors. We added a sentence to clarify that:*
*"These small errors indicate that the cameras are close to ideal cameras concerning polarization and the error introduced by using the ideal transfer matrix instead of the transfer matrices obtained from the laboratory polarization calibration is small."*
*For the DOLP, see the answer to the comment below.*

Line 390: Systematic and spatial differences between model and measurement in Figures 10 and 11 on the order of 2-6% are quite large for a flatfield residual. This may impact science retrievals done

in specific pixel regions – was there any reason not to trust the spatial distribution of the LIS field outright? Integrating spheres should be excellent spatial sources for flatfield.

The other way to approach this could be to step the specMACS field of view across the LIS aperture while taking images. This would place the LIS aperture in different locations of the FOV and fully cover the FPA in a "composite" flatfield over all images taken. Was a test like this considered? I am not requesting extra work, but for this section, I would add more details about why a model was preferred despite significant spatial residuals in Figures 10 and 11.

*Due to the large field of view, it was not possible to perform flatfield measurements covering the entire field of view. Even the composite method you propose was not possible because we could not tilt the instrument to cover e.g. also the corner regions due to its large size and weight. Because of that we chose the model to obtain a vignetting correction for the entire field of view despite the non-negligible residuals. At least part of the residuals can be attributed to inhomogeneities of the large integrating sphere. In the center of the sphere the inhomogeneities were characterized to be 0.25%. Further towards the sides the inhomogeneities are expected to be larger. On the other hand, the model does not account for pixel by pixel variations or photo response non-uniformity besides the vignetting effect which can be an explanation for some residuals.*

*We added more details about this to the text:*

*"The model was chosen for the vignetting correction despite the non-negligible residuals between the vignetting model and the flat-field measurements in order to obtain a vignetting correction for the entire field of view. Due to the large field of view of the instrument and its large size and weight, it was not possible to perform flat-field measurements covering the entire field of view of the cameras even with a composite method. The residuals include inhomogeneities of the LIS as well as deviations of the photo response non-uniformity of the cameras from the vignetting model."*

Line 435: I recommend including a table that lists the sigma errors for each of the terms in Eq. (22), for each wavelength – or if some are functions, give the functional form. Much of this data is already given throughout the paper, but a summary table is preferable.

*We tried to create a comprehensive table, however, it became very large due to the two cameras, three color channels, different Stokes vector components, and finally all components of equation 22. Because of that, instead, we added references to the respective sections where we tried to give more details and descriptions of the components of the equation where they were missing. In addition, we included the relative radiometric uncertainty into the table, to have at least the relative and absolute radiometric uncertainty given.*

Table 3. Can you also provide the uncertainty for DOLP? This is a benchmark used to gauge the overall polarization accuracy of a multi-angle polarimeter. This may take further propagation of Eq. (22), but it is also important to show (especially relative to typical atmospheric signals).

*We computed the uncertainty of the DOLP with Gaussian error propagation as suggested and added a description and discussion about it to the section. Since the DOLP is independent of the absolute radiometric response a differentiation of relative and absolute radiometric uncertainty is not reasonable and we included the DOLP uncertainties directly in the text instead of the table.*

*"Another important quantity for polarization applications is the degree of linear polarization which can be computed from the Stokes vector with DOLP = sqrt($Q^2 + U^2$)/I. The degree of linear polarization is invariant under rotations and independent of the absolute radiometric response. Its*

*relative uncertainty was computed via Gaussian error propagation from the uncertainties above. For Stokes vectors rotated into the scattering plane, the U component of the Stokes vector is much smaller than Q. Thus, neglecting the U component, the relative uncertainty of the DOLP can be calculated with $\sigma DOLP /DOLP = sqrt((\sigma I /I)^2 + (\sigma Q /Q)^2)$. It amounts to 5.4%, 5.4%, and 6.9% for the red, green, and blue channel of polLL and 4.8%, 4.9%, and 6.2% for polLR for the same typical signal level and DOLP as in Table 3. "*

---

## Author Response (AR3)

**Reply to referee comment #1**

We thank Referee #1 for reviewing the manuscript and the valuable comments and suggestions which we address below. The responses to the referee comments are given in blue italic letters.

General comments:

The paper is well written, clear, and of interest. I recommend publication with a few comments.

Specific comments:

- Eq. (1): When measuring the total intensity $I$ with a polarization camera, I think it is preferable to use $I = (I_0 + I_{45} + I_{90} + I_{135})/2$ rather than $I = I_0 + I_{90}$. Ideally of course, it should be that $I_0 + I_{90} = I_{45} + I_{135}$. Nevertheless, I think it is preferable to take the information of all four pixels into account.

  *Thank you for your comment. We do indeed compute the I-component of the Stokes vector from our measurements with $I = (I_0 + I_{45} + I_{90} + I_{135})/2$ using all four measured intensities. So, it was misleading to give the general definition of the Stokes vector in Eq. (1). We changed the equation to $I = (I_0 + I_{45} + I_{90} + I_{135})/2$ to be consistent.*

- Section 4.5.2 Laboratory polarization calibration: I think this section could be improved. I am not sure I fully understand the calibration procedure. Some additional step-by-step explanations with equations and/or a figure explaining the three reference systems and how they relate would help me. To be a bit more specific:
  - In Step 1 (Line 326-331): Did you compute Transfer matrix Eq. (8) by means of Eq. (9) with the camera being in the camera reference frame? Is the result a transfer matrix from laboratory frame (linear polarizer) to camera frame that contains a rotation matrix that still needs to be determined? Or are camera and linear polarizer in the same laboratory reference system?

    *We added more details about the different reference systems and also tried to describe the first step in more detail to make that clearer. In addition, we added a reference with sketches which visualize the different reference systems. We computed the transfer matrices in this first step in the laboratory reference frame by solving equation 9 in a least-squares sense similarly to Rodriguez et al. (2022). With the resulting transfer matrix Stokes vectors in the laboratory reference frame can be computed from measured intensities. The procedure of the laboratory polarization calibration in section 4.5.2 is independent of the theoretical polarization calibration model (equation 8) in section 4.5.1. Equation 8 gives Stokes vectors in the camera reference system. The transfer matrices obtained in the first step of the laboratory polarization calibration give results in the laboratory reference system and are rotated to the camera reference system in the second step of the laboratory polarization calibration.*

  - Line 332-345: I assume the problem that is being solved here is finding the rotation induced by the window. So if the Stokes vector is rotated beforehand, does Eq. (9) become something like this,*

$$I_n - d_n = A \cdot R \cdot S_n,$$

with R being the rotation matrix we are looking for? If so, why could you not simply fit a misalignment factor dphi similar to Eq. (13) in Lane et al. (2022) (This misalignment factor is also merely a rotation of angle dphi). I understand the sentence spanned from line 333-335, but couldn't you still optimize for the rotation by rotating the linear polarizer? How are the EURECA measurements polarized with respect to the camera reference system (or the scattering plane)? I think it is worth giving more details.

*In contrast to Lane et al. (2022) the polarizer in our setup was not mounted on a manual but on a motorized rotation stage. This means that the relative orientation of the polarizer for the different measurements with different rotation angles are very accurate and we did not have to account for misalignments due to manual arrangement. But, what we needed to determine was the absolute orientation of the 0 degree direction of the linear polarizer in camera coordinates. In principle, it would also have been possible to rotate the incoming Stokes vector first into the camera reference system and then determine the transfer matrices as you propose with your equation above. However, we did not have a "ground truth" for the 0 degree direction which we could have used to directly optimize for such a misalignment factor due to window in front of the cameras. Because of that, we used the known property, that U=0 in the scattering plane for single scattering with our measurements from the EUREC4A campaign as described in the paper. The measurements are at first raw data, from which we computed Stokes vectors in the laboratory reference frame with the transfer matrices of step 1. Then we applied two rotation matrices to the Stokes vectors, one for the transformation from laboratory to camera reference system, which had to be optimized, and the second one for transforming from the camera reference system to the scattering plane, which was known from the geometrical calibration. By minimizing U in the scattering plane we could then find the rotation from laboratory to camera reference frame.*
*We added also more details to this section to make our methods more comprehensible.*

*The entire part about the laboratory polarization calibration reads now:*

[revised manuscript text omitted]

− The statements in Line 279 "a single matrix …. To all pixels" and Line 294 "the camera lens has only little influence" citing Lane et al. are slight oversimplifications. Lane et al. used a 105m lens set to f/22 (fairly straight rays) to show that the super-pixels on the sensor are generally consistent. When they compare the lenses, they merely focus on the central pixels. However, and presumably particularly important for wide-angle lenses, lenses can show an effect called polarization aberration of lenses. This is nicely explained in the reference [1], section 1.7.2, page 22 ff. (also note the effect of high numerical aperture wavefronts described in section 1.7.3). The effect is particularly high at the edges (see Fig. 1.38 in [1]), which might explain your larger differences in the corners (mentioned in line 482). My suggestion would be: it is fair to assume one transfer matrix for all pixels, as the superpixels should generally be consistent across the entire sensor. However, this will probably not fully correct the entire lens (as you already concluded yourselves in line 486). I do not see a need to change any data / results. But it is worth to correct the statements and to mention the potential effect of polarization aberration of lenses.

*Thank you very much for pointing that out. We reworded both lines and added more details to make the differences between the setup of specMACS and Lane et al. (2022) clearer and avoid oversimplifications. Lines 279 and 294 read now:*

*"Lane et al. (2022) calibrated the monochromatic version of the polarization resolving cameras from the same manufacturer. They focused on the central pixels of the sensor and found that the transfer matrices are consistent across this sensor region and a single matrix can be applied to all pixels. In addition, the deviation between the measured matrices and the ideal matrix was small for the central pixel region with small incident angles which they considered."*

*And*

*"According to Lane et al. (2022), the choice of the camera lens has only little influence on the transfer matrices for the central pixel region of the camera where the incident angles of the rays are small. Thus, we assume that our theoretical model of the transfer matrices is a good approximation. However, lenses can introduce polarization aberrations especially for larger incident angles towards the corner regions (Chipman et al., 2018). This effect is not included in the theoretical polarization calibration model. Because of that, we validated the theoretical model with a laboratory polarization calibration."*

Technical corrections:

- Eq. (1): It should be $I_{right} - I_{left}$ and not $I_{left} - I_{right}$, see [1], page 64, Eq. (3.1)

  *Thank you very much for noting that. We corrected the equation.*

- Line 137, 142: altitude instead of attitude

  *We do indeed mean the attitude of the aircraft here. The BAHAMAS data provides aircraft position (latitude, longitude, height) and attitude (roll, pitch, and yaw angles). We added more details to clarify that:*
  *"Precise information about aircraft position (latitude, longitude, and altitude of the aircraft) and attitude (roll, pitch, and yaw angles) is available from the Basic HALO Measurement and Data System (BAHAMAS)."*

[1]: Chipman, Russell, Wai Sze Tiffany Lam, and Garam Young. Polarized light and optical systems. CRC press, 2018

**Reply to referee comment #2**

We thank Brent McBride for reviewing the manuscript and the valuable comments and suggestions which we address below. The responses to the referee comments are given in blue italic letters.

This paper discusses the calibration pipeline for the polarization resolving cameras of the spectrometer of the Munich Aerosol Scanner (specMACS). The authors discuss the instrument itself, then delve into geometric calibration of the image frame, dark characterization, non-linearity, spectral response, polarization calibration of an enclosing window and optical assembly, flatfielding, and absolute radiometric response. The authors close with a discussion of overall uncertainty and a measurement-model intercomparison over sunglint measured during a recent field campaign. This paper comes after Pörtge et al. (2023), which demonstrated the polarimetric cloud retrieval capabilities of the same specMACS cameras over marine and popcorn cumulus cloud fields. Polarimetric remote sensing is a hot topic in the climate community right now. Papers that demonstrate polarimeter instrument calibration (as well as their science) will continue to be relevant, interesting, and useful to AMT readers. I recommend publication with minor/optional revisions.

In-line comments:

Line 85: I understand that the Equation 1 is the form given in Hansen and Travis (1974), though the Sony sensor allows for a more comprehensive calculation using all four angles (shown in Lane et al. 2022):

$$I = 1/2(I_0 + I_{90} + I_{45} + I_{135})$$

This form is used later in the paper (as a normalization during polarization calibration, line 313), but it isn't clear to me if the actual intensity measurement (Stokes I) is calculated this way throughout the entire paper. Either way, please harmonize the definition of I across the paper. Also, the typical convention of V in Eq. (1) is flipped from what is shown (right − left).

*Thank you for your comment. We do indeed compute the I-component of the Stokes vector from our measurements with $I = (I_0 + I_{45} + I_{90} + I_{135})/2$ using all four measured intensities. So, it was misleading to give the general definition of the Stokes vector in Eq. (1). We changed the equation to $I = (I_0 + I_{45} + I_{90} + I_{135})/2$ to be consistent throughout the paper. In addition, we corrected the equation for the V component. Thank you very much for noting that.*

Section 3. I recommend to add a figure to this section. A visual of the chessboard calibration from the perspective of the specMACS sensor would help a lot with the interpretation here.

*We added a figure showing an example image of a chessboard as recommended.*

Line 193: Figure 3 shows that polLL and polLR have systematic differences in the forward and aft sides of the dark frame. Even at a ~2.5 counts spread, this structure is important to capture, and could be relatively easy to apply in post-processing on image data. At 30000DN, I agree it will not make much of a difference to use a single value or adopt a spatial dark map for correction. However, I imagine part of the interest in specMACS data goes beyond clouds – possibly to science retrievals

of aerosol, land, ocean, and free atmosphere properties. Many of these targets will not have 30000DN signals. For example, open ocean is extremely dark in RGB and can go to <5% in DOLP off-glint, like in Figure 12b. At these low light levels, a few counts could be important. Also, the later sections discuss the many ways that the calibration could be improved – using the spatial field of the dark (and scaling the dark counts relative to any measurement temperature) could be a step in this direction. I would consider including this in the calibration pipeline instead of a single value for the dark.

*Thank you very much for this comment. For our applications of the data to the remote sensing of cloud macro- and microphysical properties, using a single value for the dark signal is accurate enough, which is why we did not use the spatial field of the dark signal. The spread of about 2-3 counts is very small and the temperature dependency of the dark signal of about 0.16 during a typical flight negligible for our applications. But we agree that for other applications this might become relevant. We added a sentence noting that the calibration of the instrument could be improved by using the spatial field instead of single value and will keep that in mind for potential future applications to darker scenes:*
*"Moreover, the spatial field of the dark signal could be used instead of a single value for the dark signal correction to further reduce the calibration uncertainties for retrievals of e.g. aerosol or land properties with very small signal levels."*

Figure 6. It is challenging to differentiate the curves in each figure due to the overlap and large scale. This could be stronger as a residual plot (i.e. $\tilde{S_o}$ - $S_o$), as a function exposure time for all pixels shown. Also, please make the points larger.

*Thank you very much for noting that. We changed the marker and increased the size of the points as suggested to make it easier to differentiate. We are aware that the overlap of the curves in the plot is not ideal. The reason for choosing this visualization is that we wanted to show the linear scaling of the measured signal with exposure time which is not visible in a residual plot. To further quantify the deviation of the signal from the linear relationship we added the percentages in the legend of the plot.*

Line 215: Relative to the detector spec, are these non-linearities reasonable?

*The camera specifications do not include information about non-linearities. However, the non-linearities we found are reasonable compared to other cameras (e.g. Forster et al. 2020).*

Figure 7. Is there is any new information in (a) and (c) that isn't already in (b) and (d)? If not, I recommend to only show (b) and (d).

*According to Poisson statistics the variance scales linearly with the signal. This linear behavior can nicely be seen in panels (a) and (c) which is why we added those two panels. We added this to the text to make it clearer:*
*"The noise characteristics of both cameras are well captured by the Poisson model. Panels (a) and (c) show the expected linear relationship between the variance and the signal while the noise scales with the square root of the signal in panels (b) and (d)."*

Figure 8. Though the smaller peaks in blue @ 650nm and in red @ 550nm are typical of some Bayer filter designs, how is this addressed in the radiometric calibration? This could be important for cross-talk considerations, and may have some influence on the error analysis in Table 3.

*The radiometric calibration was performed at the large integrating sphere whose output spectrum was measured by the CHB. We integrated this spectrum with the previously determined spectral response functions of the three color channels to obtain the output radiance of the LIS for the respective color channels for the absolute radiometric calibration. In this sense the smaller peaks were accounted for.*

Line 278 + Line 294: This may not hold for the entire specMACS FOV, though. The Lane et al. (2022) study prioritized pixels near the image center and predicted higher errors in focus and polarization measurement at large AOI. Since a single-camera specMACS FOV is decently large (~45 deg nadir-to-aft), and the Cinegon lens does not seem to be telecentric (from the spec), there will likely be AOI-related differences in the transfer matrix at wider angles. I am glad this is recognized by the authors in the discussion towards the end of the paper, but I would reword these statements to differentiate specMACS from the Lane et al. (2022) study a bit more here.

*Thank you for this comment. We changed the lines you mentioned and added more details to make the differences between the specMACS setup and Lane et al. (2022) clearer and avoid oversimplifications. Lines 279 and 294 read now:*
*"Lane et al. (2022) calibrated the monochromatic version of the polarization resolving cameras from the same manufacturer. They focused on the central pixels of the sensor and found that the transfer matrices are consistent across this sensor region and a single matrix can be applied to all pixels. In addition, the deviation between the measured matrices and the ideal matrix was small for the central pixel region with small incident angles which they considered."*
*and*
*"According to Lane et al. (2022), the choice of the camera lens has only little influence on the transfer matrices for the central pixel region of the camera where the incident angles of the rays are small. Thus, we assume that our theoretical model of the transfer matrices is a good approximation. However, lenses can introduce polarization aberrations especially for larger incident angles towards the corner regions (Chipman et al., 2018). This effect is not included in the theoretical polarization calibration model. Because of that, we validated the theoretical model with a laboratory polarization calibration."*

Line 320: I strongly recommend to add a figure that visually explains these three reference frames. It is difficult to reconcile them from the text alone and the following paragraphs require the reader to fully understand each one.

*We added more details about the different reference systems in the text and also tried to give more detailed descriptions of the different steps of the laboratory polarization calibration in order to make our methods more comprehensible. In addition, we added a reference which includes sketches visualizing the different reference systems. The section explaining the reference systems reads now:*
*"The Stokes vector as well as the transfer matrix are always defined relative to a reference plane. In connection with the polarization calibration, we distinguish three different reference systems. The laboratory reference system is defined by the plane containing the 0◦-axis of the linear polarizer*

*between the large integrating sphere and the instrument and the normal of this polarizer. Moreover, the reference plane for the camera reference system for each camera is given by the x-z-plane of the camera coordinate system with the x-axis parallel to the 0°-direction of the polarizers on the sensor and the z-axis normal to the focal plane array of the camera. Finally, the Stokes vectors can be rotated from the camera reference system into the scattering plane. The scattering plane is the plane containing the vector of the incoming solar radiation and the viewing direction of ach pixel. Sketches visualizing the different reference systems can for example be found in Eshelman et al. (2019). The transformation from the camera coordinate system to the scattering plane is known from the geometric calibration and varies between different observation geometries with different vectors of the incoming solar radiation. Thus, with the laboratory polarization calibration, we aim for computing the transfer matrices in the camera reference system."*

Line 359: The reconstruction error on I of $10_{-13}$% is incredibly small versus the error on Q. Even with normalized intensities, I would still expect to see a reconstruction error in the ballpark of what is reported for Q. This suggests to me that the derivation of the transfer matrix is weighting the I inputs more strongly than Q or U. Can you give more details on how that value was derived?

*The transfer matrices were derived by solving equation 9 in a least-squares sense with $A = (I_n - d_n)S_n^{-1}$ where $S_n^{-1}$ is the pseudo-inverse of the incoming Stokes vectors $S_n$. The measured Stokes vectors were then reconstructed from the measured intensities via $S_{n,r} = A^{-1} (I_n - d_n)$. Finally, we computed the reconstruction error which we define as the relative difference between the reconstructed Stokes vectors $S_{n,r}$ and the incoming Stokes vectors $S_n$. The given values of the reconstruction error are mean values across all measured pixels. We added an additional reference and more detailed description of the method throughout the entire section and included the equations above to the paragraph.*

Line 365: Why is it useful to know that the specMACS pol cameras would be between 3 and 5% biased, if they were used uncalibrated while imaging a target with DOLP = 1? Most scientists will never use uncalibrated specMACS data – maybe this is a marker of how close the instrument is to an ideal calibration already? Either way, I suggest changing this to how much error we could expect to see in a calibrated specMACS DOLP measurement (or defer this to Table 3 – see comment below).

*With the laboratory polarization calibration, we analyzed the polarization properties and determined transfer matrices for significant parts of the field of view but we could not cover the entire field of view of the cameras. Because of that, we developed the theoretical polarization model and validated it with the laboratory polarization calibration. In this context, the polarization calibration error introduced by Lane et al. (2022) is useful, since it indicates that the instrument is in fact close to an ideal calibration concerning polarization. Thus, the use of the ideal transfer matrix in the theoretical model covering the entire field of view can be justified and introduces only small errors. We added a sentence to clarify that:*
*"These small errors indicate that the cameras are close to ideal cameras concerning polarization and the error introduced by using the ideal transfer matrix instead of the transfer matrices obtained from the laboratory polarization calibration is small."*
*For the DOLP, see the answer to the comment below.*

Line 390: Systematic and spatial differences between model and measurement in Figures 10 and 11 on the order of 2-6% are quite large for a flatfield residual. This may impact science retrievals done

in specific pixel regions – was there any reason not to trust the spatial distribution of the LIS field outright? Integrating spheres should be excellent spatial sources for flatfield.
The other way to approach this could be to step the specMACS field of view across the LIS aperture while taking images. This would place the LIS aperture in different locations of the FOV and fully cover the FPA in a "composite" flatfield over all images taken. Was a test like this considered? I am not requesting extra work, but for this section, I would add more details about why a model was preferred despite significant spatial residuals in Figures 10 and 11.

*Due to the large field of view, it was not possible to perform flatfield measurements covering the entire field of view. Even the composite method you propose was not possible because we could not tilt the instrument to cover e.g. also the corner regions due to its large size and weight. Because of that we chose the model to obtain a vignetting correction for the entire field of view despite the non-negligible residuals. At least part of the residuals can be attributed to inhomogeneities of the large integrating sphere. In the center of the sphere the inhomogeneities were characterized to be 0.25%. Further towards the sides the inhomogeneities are expected to be larger. On the other hand, the model does not account for pixel by pixel variations or photo response non-uniformity besides the vignetting effect which can be an explanation for some residuals.*
*We added more details about this to the text:*
*"The model was chosen for the vignetting correction despite the non-negligible residuals between the vignetting model and the flat-field measurements in order to obtain a vignetting correction for the entire field of view. Due to the large field of view of the instrument and its large size and weight, it was not possible to perform flat-field measurements covering the entire field of view of the cameras even with a composite method. The residuals include inhomogeneities of the LIS as well as deviations of the photo response non-uniformity of the cameras from the vignetting model."*

Line 435: I recommend including a table that lists the sigma errors for each of the terms in Eq. (22), for each wavelength – or if some are functions, give the functional form. Much of this data is already given throughout the paper, but a summary table is preferable.

*We tried to create a comprehensive table, however, it became very large due to the two cameras, three color channels, different Stokes vector components, and finally all components of equation 22. Because of that, instead, we added references to the respective sections where we tried to give more details and descriptions of the components of the equation where they were missing. In addition, we included the relative radiometric uncertainty into the table, to have at least the relative and absolute radiometric uncertainty given.*

Table 3. Can you also provide the uncertainty for DOLP? This is a benchmark used to gauge the overall polarization accuracy of a multi-angle polarimeter. This may take further propagation of Eq. (22), but it is also important to show (especially relative to typical atmospheric signals).

*We computed the uncertainty of the DOLP with Gaussian error propagation as suggested and added a description and discussion about it to the section. Since the DOLP is independent of the absolute radiometric response a differentiation of relative and absolute radiometric uncertainty is not reasonable and we included the DOLP uncertainties directly in the text instead of the table.*
*"Another important quantity for polarization applications is the degree of linear polarization which can be computed from the Stokes vector with DOLP = sqrt($Q^2 + U^2$)/I. The degree of linear polarization is invariant under rotations and independent of the absolute radiometric response. Its*

*relative uncertainty was computed via Gaussian error propagation from the uncertainties above. For Stokes vectors rotated into the scattering plane, the U component of the Stokes vector is much smaller than Q. Thus, neglecting the U component, the relative uncertainty of the DOLP can be calculated with $\sigma DOLP / DOLP = sqrt((\sigma I / I)^2 + (\sigma Q / Q)^2)$. It amounts to 5.4%, 5.4%, and 6.9% for the red, green, and blue channel of polLL and 4.8%, 4.9%, and 6.2% for polLR for the same typical signal level and DOLP as in Table 3. "*

**Reply to additional comments of referee #2**

We thank Brent McBride for his additional comments which we address below. The responses to the referee comments are given in blue italic letters.

However, I believe the paper still needs revisions to further clarify/justify use of the flatfield model over the LIS measurement - I am not convinced by the author response. It seems they are simplifying the calibration unnecessarily (and adding error), or the worse case - the LIS is not reliable as a uniform source.

*Due to the large field of view, it was not possible to perform flatfield measurements covering the entire field of view with the calibration setup at the CHB. Even the composite method you propose was not possible because the opening of the LIS was not large enough to cover the entire field of view at once and on the other hand we could not tilt the instrument enough to cover e.g. also the corner regions due to its large size (about 50cmx70cmx120cm) and its weight of 100kg. The calibration measurements at the LIS which we have are limited to roughly the area shown in Figures 11 b) and 12 b). Because of that we chose the model to obtain a vignetting correction for the entire field of view despite the non-negligible residuals. Part of the residuals can be attributed to inhomogeneities of the large integrating sphere. In the center of the sphere the inhomogeneities were characterized to be 0.25%. Further towards the sides the inhomogeneities are expected to be larger. In addition, the model does not account for pixel by pixel variations or photo response non-uniformity besides the vignetting effect which can be an explanation for some residuals. The standard deviation of the absolute radiometric response across all pixels in section 4.7 was below 1.4% for all channels which indicates that this effect has a minor influence. We agree that it would be much more accurate to directly use the measurements above the LIS for the calibration, but due to technical reasons it was not possible to achieve measurements for the entire field of view with the setup we had and unfortunately, we are not able to do additional measurements at the moment. We will definitely keep this in mind for future laboratory calibrations. Nevertheless, the use of the vignetting model still improves the calibration results significantly. As can be seen from Figures 11 and 12 the normalized intensities are reduced from 1.0 down to 0.8 or even lower for significant parts of the field of view. Thus, even though there are residuals up to a few percent between the model and the measurements, the model is about a factor 10 more accurate compared to using the measurements without any vignetting correction in the regions where we could not achieve any flat-field measurements.*

*We added more details about this to the text and the section reads now:*
*"Due to the large field of view of the instrument compared to the size of the LIS and the large size and weight of the instrument, it was not possible to perform flat-field measurements covering the entire field of view of the cameras with the calibration setup at the CHB. Because of that, the model was chosen for the vignetting correction in order to obtain a vignetting correction for the entire field of view despite some non-negligible residuals between the vignetting model and the flat-field measurements. The residuals include inhomogeneities of the LIS as well as deviations of the photo response non-uniformity of the cameras from the vignetting model. For future calibrations, flat-field measurements covering the entire field of view could be taken and directly be used for a more accurate flat-field correction which for example also includes pixel to pixel variations. In addition, possible inhomogeneities of the LIS could be accounted for by taking several measurements while rotating the tilted instrument above the LIS."*

Also, the calculated DOLP lab uncertainty at ~4-6% is high. This is confusing to me since the calibration matrix is close to ideal and DOLP is independent of absolute calibration. I suspect there may be an error in their derivation. Multiangle polarimeters with similar capabilities and calibration schemes can achieve < 0.5% lab DOLP error (RSP, AirMSPI, HARP etc.). The authors should revisit this calculation and clarify the discussion.

*We computed the uncertainty of the DOLP via Gaussian error propagation from the uncertainties of I and Q. For Stokes vectors rotated into the scattering plane, the U component of the Stokes vector is much smaller than Q. So, neglecting the U component, we computed the relative uncertainty of the DOLP with $\sigma DOLP / DOLP = sqrt((\sigma I / I)^2 + (\sigma Q / Q)^2)$ with the uncertainties of I and Q given in equation (22). But, since the DOLP is invariant under rotations and independent of the absolute radiometric response we did not include these uncertainties in the computation of the uncertainty of the DOLP. It amounts to 5.4%, 5.4%, and 6.9% for the red, green, and blue channel of polLL and 4.8%, 4.9%, and 6.2% for polLR for the same typical signal level and DOLP as in Table 3. These uncertainties are relatively large compared to instruments like RSP, AirHARP, or AirMSPI. However, the uncertainties given are only the upper limit and a very conservative estimate since we estimated the uncertainty of the transfer matrices with the deviation of the laboratory transfer matrices from the ideal transfer matrix using the error defined by Lane et al. (2022) as discussed in the paper. But, we included the impact of the window in the transfer matrices (see the section about the polarization calibration) which is why our transfer matrices are more accurate than the ideal transfer matrix used in the definition of the calibration error by Lane et al. (2022). In addition, Lane et al. (2022) who calibrated the monochrome version of the same sensor found maximum measurement errors for the DOLP between 3% and 8% even though they focused on the central pixel regions where the error is expected to be smaller. Thus, our results for the uncertainty of the DOLP seem to be reasonable.*
*In general, the uncertainty of the DOLP could be reduced by a more accurate laboratory (polarization) calibration. A more accurate calibration was not possible with the laboratory calibration setup at CHB given the large size and weight of specMACS as explained above. For the polarization calibration the diameter of the polarizer was small such that it only covered a small fraction of the field of view and the camera must be oriented such that the viewing directions are perpendicular to the polarizer which made it impossible for us to get polarization calibration measurements for the entire field of view. Unfortunately, we do not have access to more evolved calibration facilities for example for satellites which would potentially allow for such measurements.*

*The section about the DOLP reads now:*
*"Another important quantity for polarization applications is the degree of linear polarization which can be computed from the Stokes vector with $DOLP = sqrt(Q^2 + U^2)/I$. Its relative uncertainty was computed via Gaussian error propagation from the uncertainties above. For Stokes vectors rotated into the scattering plane, the U component of the Stokes vector is much smaller than Q. Thus, neglecting the U component, the relative uncertainty of the DOLP can be calculated with $\sigma DOLP / DOLP = sqrt((\sigma I / I)^2 + (\sigma Q / Q)^2)$. Since, the degree of linear polarization is invariant under rotations and independent of the absolute radiometric response, its uncertainty was computed from the relative radiometric uncertainties of I and Q in equation 22 neglecting the uncertainty of the absolute radiometric calibration and the rotation error. It amounts to 5.4%, 5.4%, and 6.9% for the red, green, and blue channel of polLL and 4.8%, 4.9%, and 6.2% for polLR for the same typical signal level and DOLP as in Table 3. The uncertainties of the DOLP are large compared to other polarimetric instruments like RSP, AirHARP, or AirMSPI (Knobelspiesse et al., 2019; Diner et al., 2013). However,*

*the uncertainties of the transfer matrices are a very conservative estimate as discussed above. A substantial part of this error might actually be due to the difficult calibration procedure and therefore the instrument error might be over-estimated, but we have no means to decide if this is the case. In addition, Lane et al. (2022) who calibrated the monochromatic version of the same sensor found maximum measurement errors of 3% to 8% for the DOLP even though they focused on the central pixel region where the errors are expected to be smaller. In general, the uncertainties could be reduced by a more accurate laboratory calibration with a setup that allows for taking polarization and flat-field calibration measurements for the entire field of view of the cameras."*